# Machine learning-based prediction of Alpine Foehn events using GNSS troposphere products: First results for Altdorf, Switzerland

Matthias Aichinger-Rosenberger[1], Elmar Brockmann[2], Laura Crocetti[1], Benedikt Soja[1], and Gregor Moeller[1]

[1]Institute of Geodesy and Photogrammetry, ETH Zürich, 8093 Zurich, Switzerland
[2]Swiss Federal Office of Topography (swisstopo), 3084 Wabern bei Bern, Switzerland

**Correspondence:** Matthias Aichinger-Rosenberger, maichinger@ethz.ch

**Abstract.** Remote sensing of water vapor using the Global Navigation Satellite System (GNSS) is a well-established technique and reliable data source for Numerical Weather Prediction (NWP). However, one of the phenomena rarely studied using GNSS are foehn winds. Since foehn winds are associated with significant humidity gradients between two sides of a mountain range, tropospheric estimates from GNSS are also affected by their occurrence. Time series reveal characteristic features like distinctive minima/maxima and significant decrease in correlation between the stations. However, detecting such signals becomes increasingly difficult for large data sets. Therefore, we suggest the application of machine learning algorithms for the detection and prediction of foehn events from GNSS troposphere products. This initial study develops a new, machine learning-based method for detection and prediction of foehn events at the Swiss station Altdorf, by utilizing long-term time series of high-quality GNSS troposphere products. Data from the Automated GNSS Network Switzerland (AGNES) and various GNSS sites from neighbouring countries, as well as records of an operational foehn index are used to investigate the performance of several different classification algorithms based on appropriate statistical metrics. The two best-performing algorithms are fine-tuned and tested in four dedicated experiments using different feature setups. The results are promising, especially when reprocessed GNSS products are utilized and the most dense station setup is used. Detection- and alarm-based measures reach levels between 66-80% for both tested algorithms and thus are comparable to those from studies using data from meteorological stations and NWP. For operational prediction, limitations due to the availability and quality of GNSS products in near-real time (NRT) exist. However, they might be mitigated to a significant extend by provision of additional NRT products and improved data processing in the future. Results also outline benefits for the results when including geographically relevant stations (e.g. high-altitude stations) in the utilized data sets.

## 1 Introduction

Global Navigation Satellite Systems (GNSS) are used extensively for positioning and navigation applications worldwide. Additionally, they enable users to retrieve information about the state of the Earth's atmosphere, in particular the distribution of water vapor. This technique, commonly referred to as GNSS Meteorology, was first proposed three decades ago (Bevis et al., 1992) and is still gaining increasing interest from the scientific community as well as meteorological institutions. The retrieval

of atmospheric information from GNSS is based on the fact that electromagnetic signals (such as GNSS signals) are delayed when traveling through specific layers of the atmosphere. The delay experienced by a GNSS signal in the lowest part of the atmosphere (troposphere) is proportional to the water vapor content along the signal path. This fact is typically exploited in GNSS Meteorology by introducing GNSS-derived atmospheric parameters like the Zenith Wet Delay (ZWD) or the Zenith Total Delay (ZTD) in data assimilation schemes. In numerous studies, a positive impact has been demonstrated, especially on precipitation forecasts (see e.g. de Haan (2008), Brenot et al. (2013), Bennitt and Jupp (2012), Yan et al. (2009)). However, while mostly precipitation-related studies represent the current focus of research (see Guerova et al. (2016) for a comprehensive summary), other meteorological phenomena can also be investigated by means of GNSS. The number of studies on other meteorological processes is relatively small, covering phenomena such as thunderstorm activity (de Haan (2013)) or fog formation (Stoycheva and Guerova (2015), Aichinger-Rosenberger (2018)). Stoev and Guerova (2018) represent the only investigation on foehn winds using GNSS products to our knowledge, in an initial study for Bulgaria based on observations of Integrated Water Vapor (IWV).

Foehn winds are a characteristic weather phenomenon in mountainous regions all over the world, especially in the vicinity of prominent mountain ranges like the Alps (where it is typically referred to as Alpine foehn). In general, foehn can be characterized as 'a wind (which is) warmed and dried by descent, in general on the lee side of a mountain' (WMO, 1992). This definition already includes its major characteristics observed in affected areas: strong gusty winds, increasing temperatures and decreasing humidity. While there are many other effects of foehn winds (from social to economic impacts), large wind speeds and gusts are the most critical features from the perspective of operational forecasting and warning systems. In typical foehn valleys like the Reuss Valley (Switzerland) or the Wipp Valley (Austria) wind speeds up to 100 km/h are common, and even up to 200 km/h gusts can be observed at high altitude stations.

Foehn research denotes one of the major topics of (alpine) mountain meteorology (Steinacker, 2006). Despite the fact that the underlying physical processes of foehn have been studied for over a century, still some gaps in knowledge, especially concerning small-scale features, exist. As the classical thermodynamic foehn theory is not able to sufficiently explain all observed foehn events (especially those lacking precipitation), a number of different theories and extensions have been proposed. Furthermore, large observation campaigns like the Mesoscale Alpine Program (MAP) have been conducted and combined with NWP results in order to assess small-scale effects (Gohm and Mayr (2004), Lothon et al. (2006), Drobinski et al. (2007), Mayr et al. (2007)).

Despite these substantial efforts in research, classification and forecasting of foehn both are still challenging tasks. Classification by human expertise still provides the most accurate results, as dedicated experiments, comparing subjective and objective methods, reveal (Mayr et al., 2018). The ability of NWP models to predict foehn is limited by the fact that small-scale features still cannot be modelled with sufficient accuracy due to coarse representation of real-world topography (Wilhelm, 2012).

Machine learning (ML) techniques have been a major research topic in atmospheric sciences over the last decade. ML-based approaches of post-processing NWP output, known as model output statistics (MOS), have been shown to significantly enhance operational weather forecasts, see e.g. Glahn and Lowry (1972), Wilks and Hamill (2007) or Hess (2020). ML has also been used to assign uncertainty estimates to forecasts based on deep learning methods applied to previous forecasts (Scher

and Messori, 2018). Furthermore, the classification and detection of different weather types has been advanced and automated
for different kinds of weather phenomena such as thunderstorms (Perler and Marchand (2009), Manzato (2005)), temperature
forecasts (Yalavarthi and Shashi (2009)), wind systems (Kretzschmar et al. (2004), Otero and Araneo (2021)) or large-scale
weather regimes in general (Deloncle et al., 2007). Common ML methods for such classification problems investigated in
former studies are e.g.:

- Random forests: Deloncle et al. (2007)

- Adaptive boosting (AdaBoost): Perler and Marchand (2009), Sprenger et al. (2017)

- Support vector machines : Yalavarthi and Shashi (2009)

- Neural networks: Manzato (2005), Kretzschmar et al. (2004), Otero and Araneo (2021)

Only few authors have used ML approaches for detection and prediction of Alpine foehn yet. Initial studies were carried out by
Sprenger et al. (2017), who applied the AdaBoost algorithm to a data set combining weather station observations with NWP
output fields from the Consortium for Small-scale Modeling (COSMO) model. They found good performance of the algorithm,
obtaining high values for probability of detecting foehn events (88%) and ratio for correct alarms of the algorithm (66%). The
most recent study by Mony et al. (2021) showed the feasibility of using ERA5 reanalysis and climate model output instead
of NWP output in a similar way as Sprenger et al. (2017). In addition, statistical mixture models were also applied for foehn
diagnosis, e.g. by Plavcan et al. (2014).

The presented study represents an initial investigation on the usability of GNSS troposphere product time series for the de-
tection and prediction of foehn events at the meteorological observation site Altdorf, Switzerland. Therefore we make use of
state-of-the-art ML-based classification algorithms trained and tested on a data set spanning eleven years (2010-2020), derived
from GNSS observations at sites all over Switzerland and neighbouring countries as well as a long-term record of foehn obser-
vations at Altdorf.

Although foehn diagnosis might be characterized an unsupervised learning problem (no universal truth exists and even forecasts
from human experts vary considerably (Mayr et al., 2018)), we apply supervised learning methods in this initial study. This
choice is motivated by the fact that reference data of proven quality is available (see Section 3.1.2 for details), results are easier
to interpret and our only reference study by Sprenger et al. (2017) also uses this approach. On the other hand, unsupervised
learning is typically used to cluster data in order to discover something that is not visible otherwise. For our investigation,it is
doubtful that a clear cluster would emerge solely corresponding to foehn events, considering the highly imbalanced data set
and the multitude of phenomena affecting GNSS tropospheric delays.

The average performance of different ML algorithms is assessed via a cross-validation procedure. The two best-performing
algorithms are trained over an eight-year and tested over a two-year period for different study setups (feature setups). These
feature setups cover the usage of reprocessed troposphere products as well as NRT products, which could be used for oper-
ational prediction and detection of foehn events. Furthermore, we explore the performance of an extended station network,

with the disadvantage of a shorter period of availability for algorithm training. Finally, we analyse the performance of our newly-developed method in detail over a week-long period covering two major foehn events at Altdorf.

## 2 GNSS Meteorology

As already outlined in the introduction, the concept of GNSS Meteorology is based on the fact that electromagnetic signals are delayed by the presence of the Earth's atmosphere. The signal delay is directly proportional to the refractive index $n$ of the atmosphere. In the neutral atmosphere, the refractive index or refractivity $N$ is composed of a dry ($N_d$) and wet part ($N_w$), which depend on temperature $T$ (K), as well as the dry $P_d$ (hPa) and water vapor partial pressure $e$ (hPa) respectively (Rüeger, 2002):

$$N = (n-1) \times 10^6 = N_d + N_w = \frac{77.6890 \cdot P_d}{T} + \left[ \frac{71.2952 \cdot e}{T} + \frac{3.75463 \times 10^5 \cdot e}{T^2} \right]. \tag{1}$$

The total tropospheric delay experienced by a GNSS signal observed at an elevation $el$ and azimuth direction $a$ is referred to as the Slant Total Delay (STD)

$$\text{STD}(a, el) = \text{ZHD} \cdot mf_h(el) + \text{ZWD} \cdot mf_w(el) + mf_g(el) \cdot [\text{GN} \cdot cos(a) + \text{GE} \cdot sin(a)], \tag{2}$$

where ZHD (Zenith Hydrostatic Delay) represents the hydrostatic part, and ZWD the wet part of the signal delay in the zenith direction. In addition, horizontal gradients GN (north-south direction) and GE (east-west direction), accounting for the asymmetry of the atmospheric layers passed by the signal, can be estimated in GNSS processing. In order to map the delays and gradients estimated for the zenith direction to the correct elevation, mapping functions for both parts of the delay ($mf_h(el), mf_w(el)$) and the gradients ($mf_g(el)$) are used.

The total delay in the zenith direction, i.e. the Zenith Total Delay (ZTD), is the sum of the hydrostatic and wet part

$$\text{ZTD} = \text{ZHD} + \text{ZWD}. \tag{3}$$

ZHD accounts for the major part of the total delay and is largely determined by the atmospheric pressure. It can be modelled with sufficient accuracy from surface pressure observations using, e.g., the formula of Saastamoinen (Saastamoinen, 1972):

$$\text{ZHD} = \frac{0.0022767 \cdot p_s}{1 - 0.00266 \cdot \cos(2\theta) - 0.00028 \cdot H} \tag{4}$$

where $p_s$ is the surface pressure, $\theta$ the station latitude, and $H$ is the station height above the geoid.

ZWD represents the main signal of interest for meteorological purposes, as it is directly related to the water vapor content in the air column above the GNSS receiver, and therefore to IWV, via

$$IWV = \kappa(T_m) \cdot ZWD, \tag{5}$$

where $\kappa$ denotes a semi-empirical function depending on the integrated mean temperature $T_m$. Thus, it shows the same high temporal and spatial variability as water vapor, making precise modelling from meteorological surface observations practically

impossible. As a consequence, ZWD is commonly estimated as an unknown in GNSS parameter estimation alongside of station coordinates and the receiver clock error.

## 2.1 Influence of hydrometeors on GNSS signal delays

Since foehn events can occur with and without simultaneous precipitation, the influence of hydrometeor formation on GNSS signal delays should also be described in the following. In an initial investigation over two decades ago, Solheim et al. (1999) quantified propagation delays induced in GPS signals by different types of molecular constituents such as dry air, water vapor, hydrometeors, and sand particles. They were able to show that the influence of water in both solid and liquid form on GNSS signals is significantly smaller than for its gaseous form (water vapor). However, in cases of extreme amounts of precipitation (especially in liquid form, i.e. extreme amounts of rain in very intense thunderstorms) a considerable influence for high-precision applications (as troposphere estimation) can be expected. In the framework of this study, we do not expect such severe events occurring with foehn events. Nevertheless, these events might lead to problems for our classification method (misclassification through degraded GNSS products due to high hydrometeor influence), especially in the summer months.

## 3 Methodology

### 3.1 Data

#### 3.1.1 GNSS station network and tropospheric products

All investigations presented in this study are based on GNSS troposphere products from the Automated GNSS Network Switzerland (AGNES). The AGNES network, which currently consists of 31 GNSS stations, was established in 2001 and is maintained by the Swiss Federal Office of Topography (swisstopo) (Brockmann et al., 2002). The capabilities of the network were extended to multi-GNSS in 2015 (Brockmann, 2016).

Reprocessed, long-term time series of hourly tropospheric delays and gradients, available for the period 2010-2020, are used in this study. A description of the data set as well as details on the reprocessing of GNSS data can be found in e.g. Brockmann (2015). Parts of this reprocessing were carried out in the framework of the second EUREF (International Association of Geodesy Reference Frame Sub-Commission for Europe) Permanent Network (EPN) reprocessing campaign in 2014, where GNSS data from a large number of European stations were reprocessed (Pacione et al., 2017). Therefore, also some interesting stations (from neighbouring countries such as Italy and Austria) are available and incorporated. More details on the actual selection of GNSS stations utilized for different experiments are given in Section 3.2. For this study, we use hourly GNSS troposphere products originating from reprocessing campaigns as well as operational NRT processing. The delay products are estimated every hour from 30-second measurements together with station coordinates (in this case using least-squares adjustment). Gradient products are typically estimated only every 12 hours and hourly values in between results from linear interpolation/extrapolation.

### 3.1.2 Foehn observations at Altdorf

In order to train a specific ML algorithm and evaluate its performance, a reference data set of foehn observations is needed as the target variable. This study uses time series of 10-minute estimates of foehn index (FI) calculated at the station Altdorf, following the approach presented by Dürr (2008). Altdorf is located at the exit of the Reuss Valley, between the Gotthard pass and the Lake Lucerne (see Figure 2), at a height of 449 m above mean sea level (amsl). The station has the longest time series of foehn observations in the Alps (spanning over 150 years of total observations) and FI data is provided back to 1981, for 10-minute intervals. It is also part of the National Meteorological Ground-level Monitoring Network (SwissMetNet, SMN) operated by MeteoSwiss. Currently, not only data from Altdorf but of about ten sites, frequently experiencing foehn winds, are available on an operational level. The FI introduced by Dürr (2008) is designed for operational nowcasting and relies on typical foehn predictors such as wind speed and direction, pressure and temperature gradients, and humidity observations at the respective measurement site and surrounding stations. It returns three different integer values: 0 (no foehn), 1 (foehn-mixed air) and 2 (foehn), which are distinguished based on the predictors mentioned above. In an extensive validation against classifications by human experts, the index showed good performance for indices re-calculated back to 1981 (Gutermann et al., 2012). For a detailed description of the calculation algorithm we refer to Dürr (2008) and Gutermann et al. (2012). As we aim for a binary classification (no foehn or foehn), the cases of FI = 1 are treated as nonfoehn events and therefore mapped to value 0. Furthermore, we map the cases of foehn (FI = 2) to the value 1 for the sake of simplicity in all results shown in the following. Then, each hour in the whole data set where at least one 10-minute value indicates foehn is treated as an hour of foehn appearance, and thus a foehn event.

### 3.2 GNSS station selection criteria

The final selection of GNSS stations, whose data we use as input features, is a difficult task for a number of reasons. The following list gives an overview of the main problems to keep in mind:

– Not every GNSS station provides a continuous data set of troposphere products. In fact, almost all of the available stations have data gaps over the chosen study period (2010-2020) and a large number of stations was only established after 2010.

– The ML-based detection/prediction can only be applied for foehn events where troposphere products from all selected GNSS stations (i.e. all features/predictors) are available. Since not all GNSS stations have data gaps at the same periods, the actual amount of possibly missed events can add up.

In order to document and cope with these challenges, we calculated detailed statistics for data availability for each station/feature setup used within this study, which are given in the respective result sections. Using these statistics, we try to balance the need to incorporate important stations (e.g. because of their geographic location) and still cover a sufficient amount of foehn events (for both training purposes and performance assessment). In the following, we therefore introduce the rules for the selection of GNSS stations applied in this study:

- Selected stations need to be relevant for the foehn classification, in terms of their geographical location. Based on previous studies we define an area of interest with latitude and longitude borders of [44.9°, 48.2°] and [5.9°, 10.75°] respectively, which is also shown in Figure 2.

- Time series from a selected station (of all troposphere products used) should cover at least 95% of all foehn events (hours of FI = 1) in the study period (2010-2020).

- The overall feature setup (GNSS stations and available products) should be available for at least 50% of all foehn events in the study period.

## 3.3 Feature selection from GNSS time series

The selection of features from GNSS troposphere time series is based on previous investigations on visual detection from time series of different parameters as well as on obvious choices which are expected to be impacted most by foehn conditions. Two obvious choices are visualized for December 2019 in Figure 1, namely ZWD at stations north (KALT, shown in blue) and south (LOMO, shown in red) of the Alpine ridge and its difference (bottom section, shown in black). In addition, foehn events at Altdorf are shown as color-coded periods (orange). Strong correlation between the contrary trends in ZWD at the two stations and the onset of foehn in Altdorf can be observed. Furthermore, the difference in ZWD between the stations reaches minima in the two extended foehn periods observed ($\sim$ 15.-18. and 19.-21.12.2019). These time series give a first impression how (and from which parameters) foehn events can be detected using GNSS data sets. As this becomes a very demanding task for longer periods (both visually and analytically), ML techniques are a promising tool to extend and automate such a detection process, with the additional benefit of possibly providing the ability to also predict upcoming events.

## 3.4 Default study setup

In the following the default feature setup for the algorithm comparison (cross-validation) and the first (reference) experiment is introduced. In general, the definition of a specific setup concerns the following points:

- GNSS station network: The selection of GNSS stations, from which data is utilized for training and testing the ML algorithms used, should be compliant with the criteria outlined in Section 3.2.

- Study period: In combination with the chosen station selection, a sufficient time period must be chosen to match the criteria outlined in Section 3.2.

- Features: This point defines which troposphere products should (or could) be used for this specific setup.

The detailed setup and statistics of data availability for the cross-validation and the first feature setup (FS1) are given in Table 1. It includes ZWD (absolute values and all possible differences between stations) and gradient products (GN and GE) from the chosen station network (Figure 2) as well as a selection of four ZHD differences which are representative differences between north-south stations in the network. Tests have also been conducted using all possible differences in ZHD (as for ZWD), but

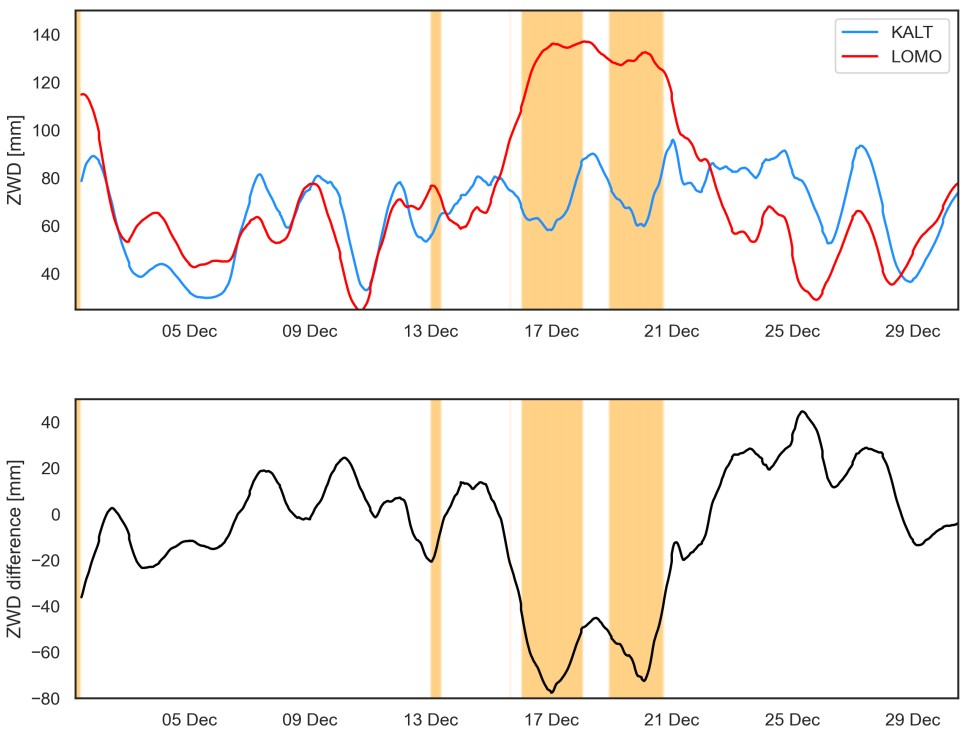

**Figure 1.** Time series of promising foehn predictors for December 2019. Top: ZWD (12-hour moving-averaged for visualization) from stations KALT (north of Alpine ridge, blue) and LOMO (south of Alpine ridge, red), Bottom: ZWD difference between KALT and LOMO (black). Observed foehn events at Altdorf (based on FI) are visualized as orange areas.

no improvement was found using this setup. This might be explained by the fact that ZHD is largely depending on pressure, which typically does not show such small-scale variations as water vapor (and thus ZWD). Therefore, a small number of ZHD

differences (i.e. pressure differences) across the Alpine ridge might be sufficient. Figure 2 provides a visualization of the full station network utilized in this study. The different colored triangles represent different station setups for the feature setups introduced later (red triangles represent stations used in the default setup, blue triangles represent stations which are added for the last feature setup). In addition, a complete list of utilized stations (including geographical coordinates) can be found in the Appendix A.

**Table 1.** Default setup used for the cross-validation and feature setup 1 (FS1). For features not outlined in a specific row, by default all stations and combinations of stations are used.

| | |
|---|---|
| Training period | 2010-2018 |
| Test period | 2019-2020 |
| Station setup | Post-processed: red triangles in Figure 2 |
| Feature setup | ZWD, GN, GE, ZWD_diff, ZHD_diff |
| ZHD_diff combinations | KALT-STA2, LUZE-STA2, BOU2-STA2, SIGM-TORI, ETHZ-TORI |
| Total number of features | 564 |
| Foehn events (FI == 1) | 5642 hours |
| Foehn events without GNSS data | 2049 hours (36.3%) |

## 3.5 Data preparation

One of the main challenges for ML-based classification algorithms are imbalanced data sets. This imbalance is also strongly present in data sets of foehn observations, since foehn is a rather rare meteorological phenomenon. For the utilized FI data set, the average foehn probability over the 11-year period (2010-2020) amounts to only $\sim 4\%$. Thus, the ratio of under-representation of the minority class (foehn event) compared to the majority class (no foehn event) is as large as 1:25.

### 3.5.1 Oversampling

A common approach to overcome problems originating from high imbalance in a data set is to oversample the minority class for the training data set. One possible approach to achieve this is the Synthetic Minority Over-sampling Technique (SMOTE) (Chawla et al., 2002), which we use in this study. The technique creates new (synthetic) instances of the minority class within the training data. For this study, an oversampling of observed foehn hours in the training data set by 25% was conducted using SMOTE, which improves the performance of the applied algorithms by about 20%. The value of 25% oversampling was chosen to achieve a reasonable balance between the advantage of having more usable training events (larger percentage of oversampling) and the fact that foehn is still a rather rare phenomenon (therefore also rare in possible test data sets). All results shown in the following sections are based on pre-processing using this approach.

### 3.5.2 Shifting of FI time series

In order to assess the suitability of the GNSS troposphere products for operational prediction, a time shift of one hour is applied to the target vector (i.e. FI time series at Altdorf). As operational usage is considered a future goal of the proposed method, the shift is applied for all feature setups investigated in this study (also those using post-processed GNSS products). Therefore, each prediction of a foehn event is based on GNSS observations collected one hour before a possible onset of foehn at Altdorf.

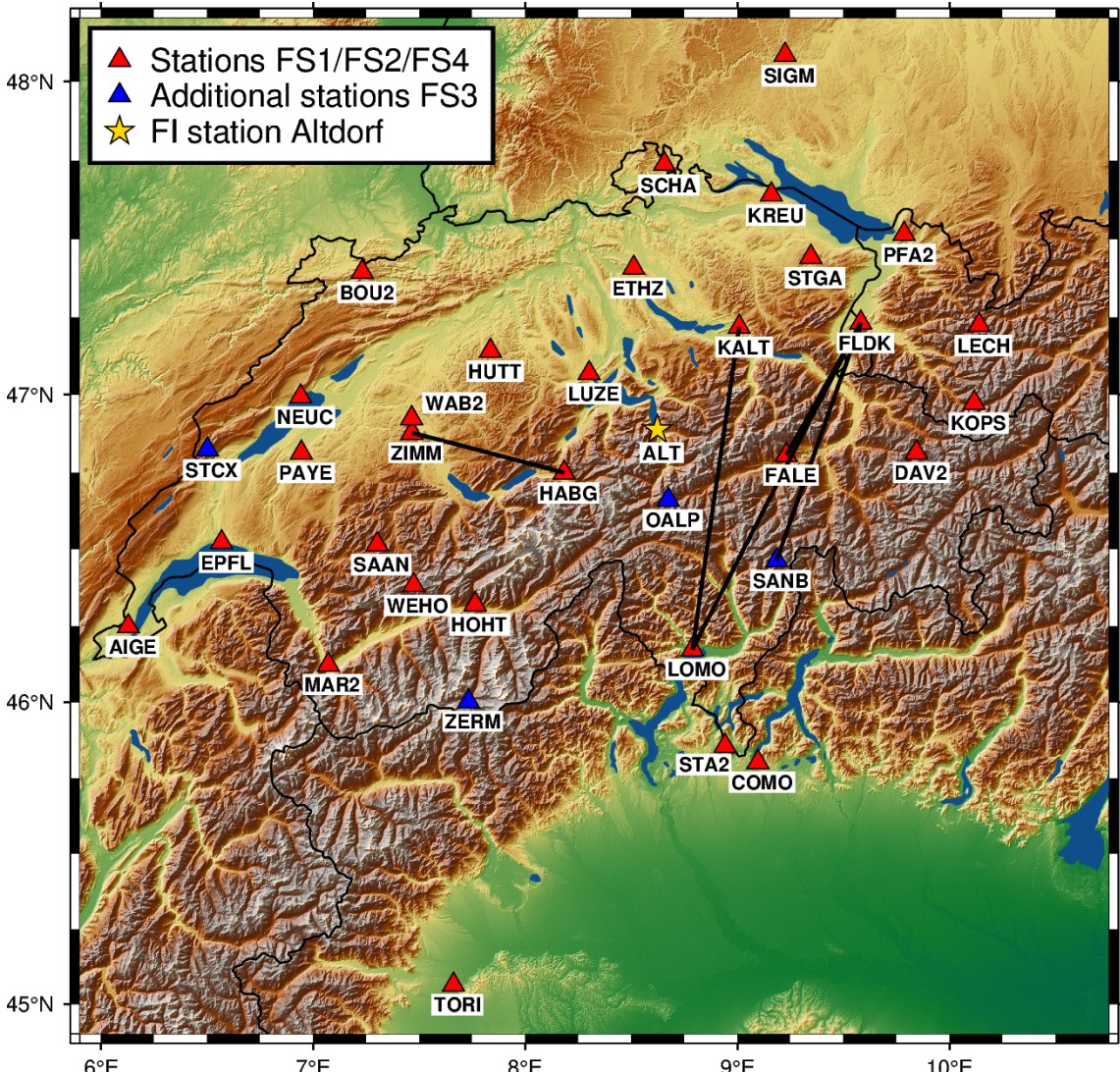

**Figure 2.** GNSS station network utilized for this study. Stations included in the default setup (cross-validation and all feature setups) are marked as red triangles. Stations explicitly used in Feature setup 3 (FS3) are marked as blue triangles. The SMN-station Altdorf is marked with a yellow star. In addition, the black lines represent the ZWD differences between stations which serve as the top predictors in FS3, shown in Section 4.3.

## 3.6 Performance metrics

As already outlined in the last section, the imbalance in data sets of foehn observations is a major obstacle for the application of ML algorithms and the assessment of their performance. For highly imbalanced data, performance metrics typically used in ML studies might not be representative and therefore other options have to be explored. In the case of the present data set, a typical

performance measure such as precision alone would not be suitable as it simply compares detected/predicted foehn hours with the observed data for all time steps. Thus, it might happen that an algorithm with optimal precision predicts (almost) no foehn

events at all, since this will still result in an optimal performance with regards to precision. In order to overcome these issues we adapt the following performance metrics (see e.g. Barnes et al. (2007)), the same ones as in Sprenger et al. (2017). These can be formulated as conditional probabilities P(|) and calculated using the so-called confusion matrix. The matrix reports the number of false negatives (FNs), false positives (FPs), true negatives (TNs), and true positives (TPs) and thus allows for the calculation of common performance measures in ML such as precision, recall.

The measures used in this study can be separated into detection-based:

- Probability of Detection (POD) = P(predicted | observed)

- Probability of False Detection (POFD) = P(predicted | not observed)

- Missing Rate (MR) = P(not predicted | observed)

and alarm-based metrics:

- Correct Alarm Ratio (CAR) = P(observed | predicted)

- False Alarm Ratio (FAR) = P(not observed | predicted)

- Missing Alarm Rate (MAR) = P(observed | not predicted).

As already visible from the formulations above, POD and CAR are directly connected to each other via the Bayes Theorem. This also implies that there is always a trade-off between those two parameters and therefore only one of them can be optimized,

while decreasing the respective other one. Which metric should be optimized strongly depends on the actual application, as already outlined by Sprenger et al. (2017), who argued that alarm-based measures might be more relevant from a forecaster's perspective.

In addition, we adopt two measures which represent a kind of mean performance in terms of both CAR and POD for describing the results presented in the next sections. The first one is just the simple average of those two parameters combined, therefore

referred to as COMB in the following:

$$\text{COMB} = \frac{\text{POD} + \text{CAR}}{2}. \tag{6}$$

The second adopted metric is based on the F-beta score $F_\beta$ (Baeza-Yates and Ribeiro-Neto, 1999), which can also be formulated using the confusion matrix. Using precision and recall measures, the $F_\beta$ score can be computed for varying $\beta$:

$$F_\beta = (1 + \beta^2) \cdot \frac{\text{precision} \cdot \text{recall}}{(\beta^2 \cdot \text{precision}) + \text{recall}} \tag{7}$$

The classical F-beta score ($F_1$, using $\beta = 1$) represents the weighted harmonic mean of precision and recall, with a range between 0 (worst case) and 1 (optimal value). As already discussed above, a precision measure might not be representative for

results of this study as this might result in our algorithm predicting no foehn events at all. Thus, we use the F2 score (beta = 2) which weights the recall measure two times larger than precision measure, instead of $F_1$ in order to put more emphasis on the recall, i.e. on the detection of all foehn events:

$$F_2 = 5 \cdot \frac{\text{precision} \cdot \text{recall}}{(4 \cdot \text{precision}) + \text{recall}}.$$
(8)

## 3.7 Algorithm selection and tuning

This section gives an overview on tested algorithms and the process of algorithm selection using a cross-validation approach. Details on this approach are given in the following Section 3.7.1. Furthermore Section 3.7.2 presents the means how the chosen algorithms are optimized via a grid search procedure.

### 3.7.1 Machine learning algorithms

In the course of this study, several different ML algorithms are tested in order to investigate their usability for this specific problem and to compare their performance relative to each other. The following algorithms, which have already been applied for classification of foehn or other meteorological phenomena (as listed in the introduction), are tested:

- Adaptive Boosting (AdaBoost) (Freund and Schapire, 1997)

- Gradient Boosting (GB) (Friedman, 2001)

- Multilayer Perceptron (MLP) (LeCun et al., 2012)

- Random Forest Classifier (RF) (Breiman, 2001)

- Support Vector Classifier (SVC) (Platt, 1999)

- K-Nearest Neighbor (KNN) (Cover and Hart, 1967)

As a detailed discussion of all algorithms would go beyond the scope of this study, we focus on the chosen ones at the end of this section. For a comprehensive overview of all of them, we refer to Hsieh (2009).

Before carrying out experiments using a specific ML algorithm, the most promising one(s) have to be identified from the list of algorithms given above. Therefore, a cross-validation over the training data set (2010 - 2018) was performed and evaluated using the performance metrics introduced in Section 3.6. For the cross-validation, single years of data are iteratively taken out
of the training data set, serving as validation data in order to assess the performance of the outlined algorithms. This is repeated until every year serves once as validation data set. The actual implementation is carried out using the Python package scikit-learn (version 1.1.2) (Pedregosa et al., 2011). For these cross-validation runs, the default settings from the algorithm routines are used in order to get an objective, initial picture of their performance for this problem. More sophisticated approaches like including more (complex) algorithms or running grid searches for each algorithm separately before the comparison were not
followed due to the "proof-of-concept" focus of the study. However, these approaches might be used in future studies on this

topic, in order to optimize the overall performance (see also the future ideas described in the outlook (Section 6)).

Resulting statistics of the cross-validation are summarized in Table 2. These results indicate best performance for the SVC algorithm in terms of combined measures (COMB and $F_2$ score). For detection-based measures (POD), the KNN algorithm achieves the highest value on average but falls short in terms of CAR and POFD. The same holds for the RF algorithm in terms of alarm-based measures (CAR), but its detection-based performance is significantly degraded compared to e.g SVC. Ultimately, we decided to use the GB and SVC algorithms for evaluation in the feature setup experiments, as these are the only ones providing average combined measures of over 70% (see Table 2). Based on the results of the cross-validation, the GB

**Table 2.** Averaged performance metrics of the cross-validation for all tested algorithms over the nine year training period 2010-2018.

| Algorithm | POD | CAR | COMB | $F_2$ | POFD | MAR |
|---|---|---|---|---|---|---|
| AdaBoost | 0.575 | 0.504 | 0.540 | 0.559 | 0.039 | 0.030 |
| GB | 0.757 | 0.664 | 0.711 | 0.736 | 0.027 | 0.016 |
| MLP | 0.702 | 0.686 | 0.694 | 0.696 | 0.024 | 0.021 |
| RF | 0.579 | 0.789 | 0.684 | 0.610 | 0.011 | 0.029 |
| SVC | 0.764 | 0.721 | 0.742 | 0.754 | 0.021 | 0.016 |
| KNN | 0.791 | 0.407 | 0.599 | 0.665 | 0.080 | 0.015 |

and SVC algorithm are chosen for the experiments shown in Sections 4 and 5. In the following we give a brief introduction of those two algorithms.

1. Gradient Boosting (GB): Introduced by Friedman (2001), GB represents a technique of ensemble learning. It builds a prediction model by an additive combination of weak learners. Typically decision trees, which are built sequentially in an iterative manner, are used. GB represents a supervised model which can deal with both classification and regression problems. The main hyperparameters of the algorithm (tuned in the next section) are:

   – n_estimators:number of boosting stages to perform

   – max_depth: maximum depth of the individual regression estimators that limits the number of nodes in the tree

   – learning_rate: limits the contribution of each tree

2. Support Vector Classifier (SVC): The SVC (Platt, 1999) also denotes a supervised model which classifies samples by searching for the best hyperplane separating data points of one class from those of the other class. The basic version of SVC is a linear classifier, finding the best linear separation between two classes. In order to solve non-linear problems, one can make use of the kernel formulation, which uses a radial basis function (RBF) in our study. Through the combination of several binary classifiers, binary problems can be extended to multiclass classification.
   The main hyperparameters for SVC are:

   – C: regularization parameter

– gamma: represents the kernel coefficient

– kernel: kernel type to be used in the algorithm, set to radial basis function (RBF) in this study.

### 3.7.2 Hyperparameter tuning

In order to optimize the performance of the chosen algorithms, dedicated tuning of their hyperparameters is carried out. Therefore, a (small-scale) grid search procedure is conducted, which is an exhaustive search over a subset of manually selected values. The performance of all hyperparameter value combinations is evaluated based on a three-fold cross-validation. Therefore, the training data set (2010-2018) is randomly divided into three folds, where two thirds are used for training while the last third serves for validation. This procedure is repeated three times until each third is used once for validation. All tested hyperparameter values, as well as the best performing value combinations, are summarized in Table 3. We waive to do a

**Table 3.** Tuned hyperparameters of both algorithms with their best, tested and default values.

| Algorithm | Hyperparameter | Best Value | Tested Values | Default Value |
|---|---|---|---|---|
| GB | n_estimators | 300 | [100, 300, 500] | 100 |
| | max_depth | 5 | [3, 5, 8] | 3 |
| | learning_rate | 0.1 | [0.05, 0.1, 0.2] | 0.1 |
| SVC | C | 0.1 | [0.1, 1, 10, 100, 1000] | 1 |
| | gamma | 'scale' | [1, 0.1, 0.01, 0.001, 0.0001,'scale'] | 'scale' |

more intensive grid search procedure for reasons already mentioned in the cross-validation section ("proof-of-concept" study), although we plan to optimize the performance of the method through this in future studies.

## 4  Results: Feature setup experiments

As the major performance test of the proposed method, four experiments are performed. Within these experiments, different setups regarding utilized GNSS stations and tropospheric parameters in the feature matrix are investigated.

### 4.1  Feature setup 1: Reprocessed products

Feature setup 1 (FS1) investigates the performance of the chosen algorithms for an optimal combination of the largest station network and longest study period, still compliant to the selection criteria from Section 3.2. For this setup, the reprocessed time series of GNSS troposphere products are used, thus also tropospheric gradients can be utilized.

Resulting statistics for FS1 from both algorithms are given in Table 4. Both algorithms show a promising performance, which, on average, is comparable to the results reported by Sprenger et al. (2017). For the GB algorithm, POD values significantly (13%) lower than for the reference study, are balanced by an improvement in CAR by $\sim 10\%$. The SVC algorithm tends

to predict more events (higher POD) and therefore also produces more false-alarms (higher POFD, lower CAR), somewhat similar to the AdaBoost algorithm of Sprenger et al. (2017). Combined measures (COMB and $F_2$) indicate very similar overall performance for both tested algorithms, with GB having the advantage of more balanced (almost equal) values for detection- and alarm-based measures. Furthermore, the actual probability of foehn events predicted by both algorithms over the test period (3.7/4%), lies within one percentage point of the actual observed probability (4.7%) for all events where GNSS-based results could be generated. Table 8 provides the confusion matrix statistics of the GB and SVC algorithm for all setups investigated

**Table 4.** Performance metrics for the proposed models using post-processed troposphere products and the full feature setup (as shown in Table 1).

| Algorithm | POD | CAR | COMB | $F_2$ | POFD | MAR | P_predicted | P_observed |
|---|---|---|---|---|---|---|---|---|
| GB | 0.753 | 0.764 | 0.758 | 0.7555 | 0.011 | 0.012 | 0.037 | 0.047 |
| SVC | 0.804 | 0.663 | 0.733 | 0.771 | 0.020 | 0.010 | 0.040 | 0.047 |

in this study. The uppermost two lines provide the values of TN, FP, FN and TP for FS1. First of all, the results show the large imbalance of the classification problem, with TN values being approximately 30 times larger than TP values. Furthermore it can be seen, that SVC produces a significantly larger amount of FP values (false alarms), which is also reflected in the statistics given in Table 4. On the other hand, it also misses less events compared to the GB algorithm, resulting in a lower number of FNs and thus lower MAR. GB shows an approximately equal number of FP and FN values, but providing more TN values than SVC.

In addition to the statistics provided in Table 4 and 8, Figure 6 shows observed (red) and predicted (orange=GB and blue=SVC) foehn events for FS1 (top left corner), as well as the results for all other FS experiments. It should be noted that all observed events are shown, including those where no GNSS data is available and thus no prediction could be made. This fact also has to be kept in mind when looking at the results in Table 4, otherwise they might look to optimistic. Nevertheless, some overall conclusions such as SVC predicting more events than GB (higher POD but lower CAR) is still visible from Figure 6.

Another major advantage of the GB algorithm is its ability to assess the importance of the used features for the prediction result. In Figure 3, we show the 20 most important predictors (features) for the classification carried out for FS1. By far the best predictor is the ZWD difference between the stations FLDK and FALE, which is surprising due to the fact that both stations are not as close to Altdorf as others in the utilized network. Interestingly, also features from stations even further away from Altdorf have significant impact, most prominently ZWD and also gradient products (even for east-west direction), e.g. in the Valais area (WEHO and HOH2 stations). This is reasonable due to the fact that typical wind trajectory in the Rhone valley is east-west oriented. ZHD differences, representing larger-scale pressure gradients (such as LUZE-STA2 or ETHZ-TORI) are also found in Figure 3. This is consistent with the study from Sprenger et al. (2017), where the pressure difference between Zürich and Locarno was the single most important predictor.

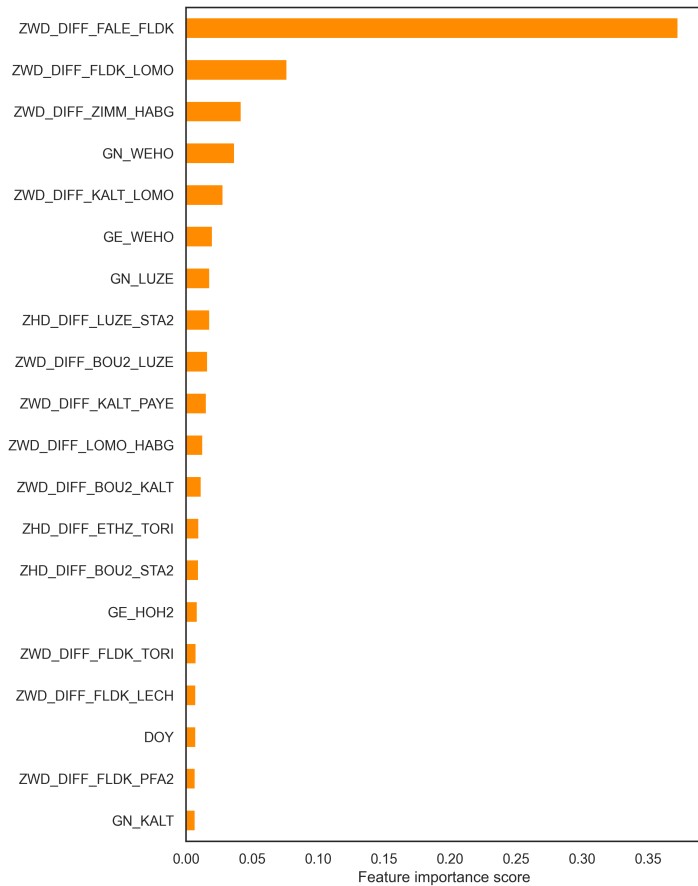

**Figure 3.** Feature importance score of the 20 top predictors for the GB algorithm for the setup used in FS1.

## 4.2 Feature setup 2: NRT products

Feature setup 2 (FS2) addresses the major question to what degree the proposed method can be used for (or incorporated into) operational forecasting of foehn events. Therefore, the investigations presented for FS1 are extended by using NRT troposphere products for the two-year test period. This way, we investigate the suitability of the proposed ML algorithms for operational prediction. NRT troposphere products are currently available in form of tropospheric delays (ZHD, ZWD and ZTD), which are typically provided with a latency of approximately 30-40 minutes after each full hour. Unfortunately, no atmospheric gradients are currently delivered in NRT mode, but an extension is possible and aimed for in the near future. The missing gradient information makes it necessary to train the GB and SVC algorithms again for the dedicated period (2010-2018 of reprocessed products), but this time only using features related to tropospheric delays (ZWD, ZTD, ZWD differences, ZHD differences). The detailed setup used for FS2 is again aligned with the criteria outlined before and given in Table 5. Resulting statistics of the prediction using NRT products can be found in Table 6. In comparison to FS1, a general performance decrease in all

**Table 5.** Default setup used for FS2. For features not outlined in a specific row, by default all stations and combinations of stations are used.

| | |
|---|---|
| Training period | 2010-2018 |
| Test period | 2019-2020 |
| Station setup | Default network: Red triangles in Figure 2 |
| Feature setup | ZWD, ZTD, ZWD_diff, ZHD_diff |
| ZHD_diff combinations | KALT-STA2, LUZE-STA2, BOU2-STA2, SIGM-TORI, ETHZ-TORI |
| Total number of features | 501 |
| Foehn events (FI == 1) | 5642 hours |
| Foehn events without GNSS data | 2004 hours (35.5%) |

measures is apparent for both algorithms. For the SVC algorithm, the degradation is more pronounced for the alarm-based ($\sim$ -6% in CAR) than for detection-based measures (almost equal performance). The GB algorithm shows equal degradation for both types. Rows three and four of Table 8 give the confusion matrix entries for FS2. Results for the different algorithms are

**Table 6.** Performance metrics for FS2 using NRT GNSS products.

| Algorithm | POD | CAR | COMB | $F_2$ | POFD | MAR | P_predicted | P_observed |
|---|---|---|---|---|---|---|---|---|
| GB | 0.694 | 0.693 | 0.694 | 0.694 | 0.014 | 0.014 | 0.0428 | 0.0432 |
| SVC | 0.799 | 0.612 | 0.706 | 0.753 | 0.023 | 0.010 | 0.0564 | 0.0432 |

almost identical to those of FS1. As for FS1, SVC produces a larger amount of FP values, but misses less events compared to the GB algorithm. GB again shows an approximately equal number of FP and FN values and provides more TP values.

The top right corner of Figure 6 shows observed vs. predicted foehn events for FS2, in a similar way as already described for FS1. It also confirms the conclusions drawn from statistics given in Tables 6 and 8, namely that using NRT products increases the number of false alarms (lowering CAR) compared to post-processed data (FS1). This holds true especially for the GB

algorithm, where significantly more predicted events are visible for FS2 compared to FS1.

This indicates the importance of gradient parameters for the proposed method, as already visible in feature importances of FS1 (Figure 3). Furthermore, lower quality of ZWD estimates must be taken into consideration as well for the NRT solution, since lower-quality orbit and clock products have to be used for GNSS processing. Nevertheless, combined measures ($F_2$ and COMB) are still reaching values around 70%, which might qualify the method as a potentially beneficial additional tool for

operational forecasting, if efficient mitigation strategies for missing GNSS data can be applied. Some possible strategies will be discussed in the outlook section (Section 6).

In the absence of gradient parameters, ZWD differences dominate the top 20 of most important features of the GB algorithm as to be expected (see Figure 4). The dominance of the top predictor (again ZWD difference between FALE and FLDK) is even

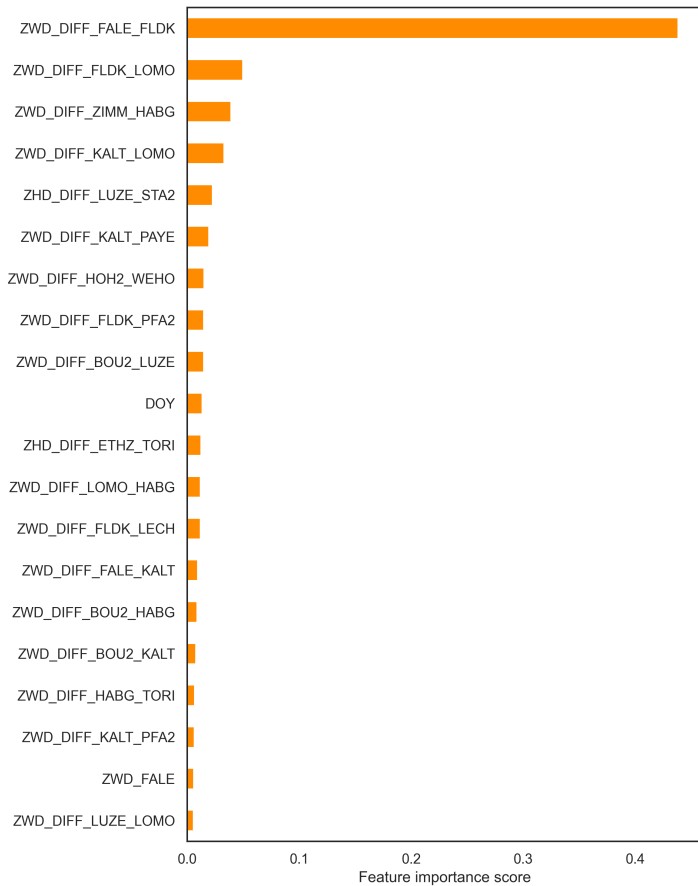

**Figure 4.** Feature importance score of the 20 top predictors for the GB algorithm for FS2.

more pronounced than for FS1. Nevertheless, also ZHD differences and DOY are again present and absolute ZWD for certain
stations also shows up.

### 4.3 Feature setup 3 and 4: Shorter period/more stations

This additional experiments investigate the question if the time period of training data can be reduced when at the same
time new stations (and therefore features) are introduced. This is especially important since it allows for the introduction of
geographically interesting stations (e.g. located at higher altitudes such as SANB or OALP, see Appendix A for more details),
which might be beneficial for the algorithm performance. The details of the feature setup for both experiments are given in
Table 7. The utilized GNSS parameters remain unchanged from the default setup, but in addition four stations (marked in blue
in Figure 2) are added to the station setup. The study period is adjusted to 2015-2019 in order to again be aligned with the
selection criteria and this period is split into training and test period by 80/20% proportion.

In order to make a direct comparison possible, a fourth feature setup (FS4) is introduced in addition. It uses the station setup of the former experiments (FS1 and FS2) but the study period of FS3. Resulting statistics for both experiments (FS3 and FS4) are given in Table 9. The results for FS3 show improvements for both algorithms, especially for alarm-based measures,

**Table 7.** Default setup used for FS3/FS4. For features not outlined in a specific row, by default all stations and combinations of stations are used.

| | |
|---|---|
| Training period | 2015-2018 |
| Test period | 2019 |
| Station setup | Post-processed: Red+blue / red triangles in Figure 2 |
| Feature setup | ZWD, GN, GE, ZWD_diff, ZHD_diff |
| ZHD_diff combinations | KALT-STA2, LUZE-STA2, BOU2-STA2, SIGM-TORI, ETHZ-TORI |
| Total number of features | 738/564 |
| Foehn events (FI == 1) | 2650 hours |
| Foehn events without GNSS data | 1244 hours (46.9%) / 745 hours (28.1%) |

compared to FS4. This outlines the importance of including geographically relevant stations (especially stations at the crest, i.e. higher altitudes) for the performance of the methods. Confusion matrix elements for FS3 and FS4 can be found in the lower half of Table 8. Overall, they show the same behaviour as already observed for the other experiments. However, it should be noted that e.g. the relative amount of FNs is lower for this feature setup, which is also reflected in the (improved) resulting statistics. One major reason for this is the higher probability of foehn events during 2019 in general. This is evident from both Table 9 (probability only considering events which are also covered by GNSS data) as well as from Figure 6 (showing all observed events). Another reason is that introducing new stations also leads to a significant decrease in foehn events, for which results are available. This is clearly visible by comparing time series for FS3 and FS4, e.g. for the GB algorithm in November 2019 (around DOY 315-320). Still, benefits can be attributed of the newly introduced stations, as they are also reflected in the feature importances of the GB algorithm, visualized in Figure 5. In this case, the top predictor of FS1 and FS2 (ZWD difference between FLDK and FALE) is superseded by the ZWD difference between FLDK and SANB (station at San Bernadino pass, 1702 masl). The station SANB, as well as other high-altitude stations (such as OALP), are present a few times in the top 20 for FS3. ZHD differences are actually no longer among the top predictors for FS3, which might be explained by the fact that ZWD observations (and having them at relevant location) still provides significantly more valuable information. Feature importances for FS4 are not shown as they are excepted to be fairly similar to those obtained for FS1, for which the same station setup and products are used. Overall, it can be concluded that benefits from additional stations can be excepted, although for future applications of the method it will be crucial to ensure data availability, especially for the top predictors shown in Figure 5.

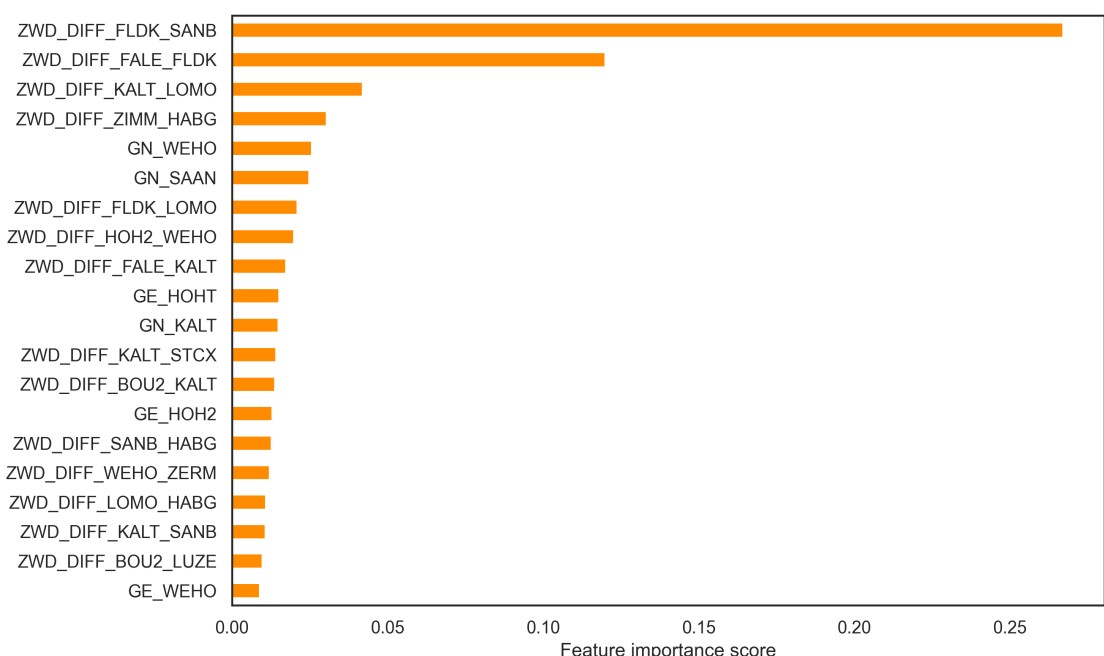

**Figure 5.** Feature importance score of the 20 strongest predictors for the GB algorithm, when using the FS3 setup.

**Table 8.** Confusion matrix statistics for all four feature setup experiments.

| Setup | Algorithm | TP | TN | FP | FN |
|-------|-----------|------|------|-----|-----|
| FS1 | GB | 324 | 8643 | 100 | 106 |
|     | SVC | 346 | 8567 | 176 | 84 |
| FS2 | GB | 277 | 8714 | 123 | 122 |
|     | SVC | 319 | 8635 | 202 | 80 |
| FS3 | GB | 222 | 3797 | 57 | 71 |
|     | SVC | 249 | 3778 | 76 | 44 |
| FS4 | GB | 258 | 4249 | 71 | 105 |
|     | SVC | 294 | 4214 | 106 | 69 |

## 5 Results: Performance analysis for distinctive foehn events

The performance statistics shown in the last section give an overview of the average performance over the test period. In order to gain more insight on the performance of our method for specific events, we look at a week-long period (including two foehn events) at Altdorf in the year 2019 in more detail. Therefore we make use of results from both algorithms for FS3, and evaluate

**Table 9.** Performance metrics for FS3 and FS4.

| Setup | Algorithm | POD | CAR | COMB | $F_2$ | POFD | MAR | P_predicted | P_observed |
|-------|-----------|-----|-----|------|-------|------|-----|-------------|------------|
| FS3 | GB | 0.758 | 0.796 | 0.777 | 0.765 | 0.015 | 0.018 | 0.065 | 0.071 |
| | SVC | 0.850 | 0.766 | 0.808 | 0.832 | 0.019 | 0.012 | 0.078 | 0.071 |
| FS4 | GB | 0.711 | 0.784 | 0.747 | 0.724 | 0.016 | 0.024 | 0.070 | 0.078 |
| | SVC | 0.810 | 0.735 | 0.772 | 0.794 | 0.025 | 0.016 | 0.070 | 0.078 |

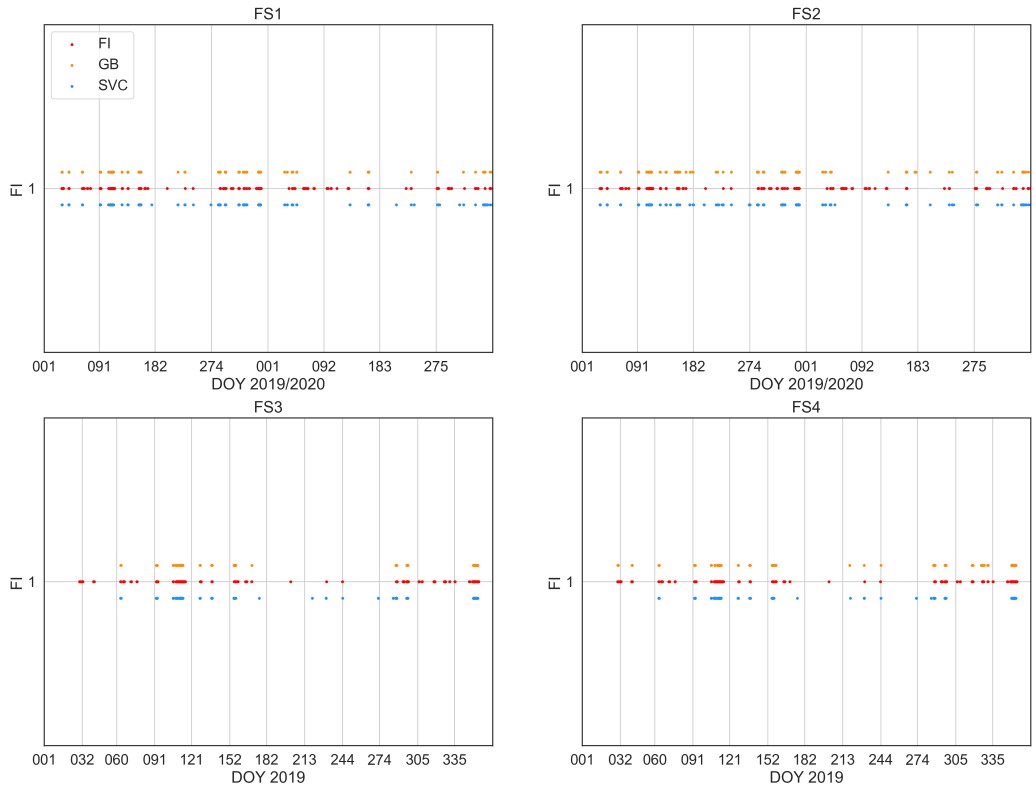

**Figure 6.** Time series of observed (red) and predicted (orange = GB, blue = SVC) foehn events for all FS experiments and their respective time periods (2019-2020 or 2019). Note that (1): curves for GB/SVC are shifted by -0.01/+0.01 with respect to the observed events (FI == 1) and (2): all observed foehn events are shown here, also those for which no prediction could be made due to missing GNSS data.

their performance for this specific week against the operational FI.

Between 15.-21.12.2019, two major (south) foehn events were observed at Altdorf, for which we analyse the performance of
our method in the following. Relevant meteorological parameters describing the situation are visualized in the upper two panels
of Figure 7. The upper part shows observations of temperature (red) and relative humidity (green) as well as wind speed (black)
and direction (grey) for the time period 15.-21.12.2019 (DOY 349-355). The first major event started around midnight at DOY
350 (16.12.2019), as visible in significant increase/decrease in temperature/relative humidity as well as the onset of a strong
southerly flow (up to 60 km/h). This situation persisted over 48 hours, until the early morning hours of DOY 352 (18.12.2019).
After a short break during daytime, the second major event started in the late evening of that same day and again lasted close to
48 hours until the evening of DOY 354 (20.12.2019), again accompanied by similar conditions observed at the SMN station.

The bottom part of Figure 7 shows the classification results of our method for GB (orange) and SVC (blue) algorithm as well
as the reference FI index (red) from MeteoSwiss. First of all, it must be noted that for both the initial hours of the first major
event and the last hours of the second event, no full feature matrix was available and therefore no classification was possible.
This highlights the major drawback/limitation of the introduced method. However, as soon as all necessary data is available,
the event is captured by both algorithms.

An overall assessment of the results shows a good average performance of the GNSS-based classification, especially for the
GB algorithm. The SVC algorithm tends to over-predict foehn during the break period and therefore issues a large number of
false alarms. This fact is also reflected in the statistics of all feature setups shown in Section 4 (higher POD but also higher
POFD/lower CAR).

However, the transition from foehn to non-foehn and vice versa is not as accurately captured as in the operational FI index.
One possible reason might be a slightly longer (or shorter) response time of the GNSS-based parameters at a specific station
to a change in synoptic conditions, relative to the classical meteorological parameters observed at Altdorf. Therefore, it might
be beneficial to introduce different time lags on some of the predictors depending on the actual physical parameter and the
geographical location of the contributing stations.

In order to analyse the performance of the methods in more detail, and increase understanding of the physical processes
captured in the GNSS products, we additionally analyse the time series of the most important predictors (features) of the in-
troduced method. Figure 8 shows the six top predictors of the GB classification for FS3, as shown in Figure 5 as well as the
observed foehn periods (color-coded in red). Most of them show distinctive patterns for foehn as well as non-foehn periods,
which are consistent with the physical relationships between GNSS observations and meteorological conditions (as introduced
in Sections 2 and 3.3) experienced for the respective period. ZWD differences show local minima or maxima for the major
foehn periods, as already suspected in Section 3.3. Whether minima or maxima are observed depends on the actual location
of the stations and on how the difference is built. Most of the (ZWD difference) features shown in Figure 8 are built as a
north-south station difference, which was expected to be a good predictor for foehn. Typically, low ZWD values are observed
at stations north of the Alpine ridge (such as FLDK here) and high ZWD values at stations south of the Alpine ridge (e.g.
LOMO). The north gradient observed at station WEHO (GN_WEHO) is also consistent with the physical understanding,

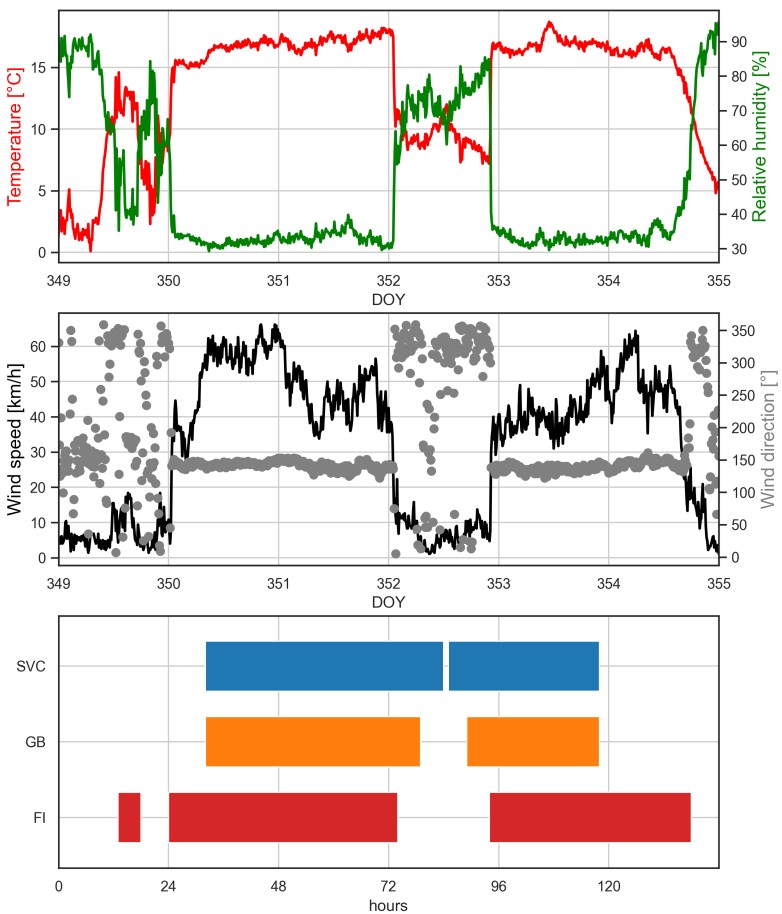

**Figure 7.** Time series of meteorological observations (top: temperature (red) and relative humidity (green), middle: wind speed (black) and direction (grey)). Bottom: classification results of SVC (blue) and GB (orange) and the reference FI index (red)

showing a southward (negative) trend for the observed foehn periods. The only feature which might not be intuitive is the ZWD difference between the stations ZIMM (near Bern) and HABG (located a bit north of the main Alpine rigde). Although
the extreme values are not as pronounced as for other predictors, they are still visible in the shown time series. This suggests that in the majority of cases, humid air reaches also the station HABG, which is located further north of the main Alpine ridge.

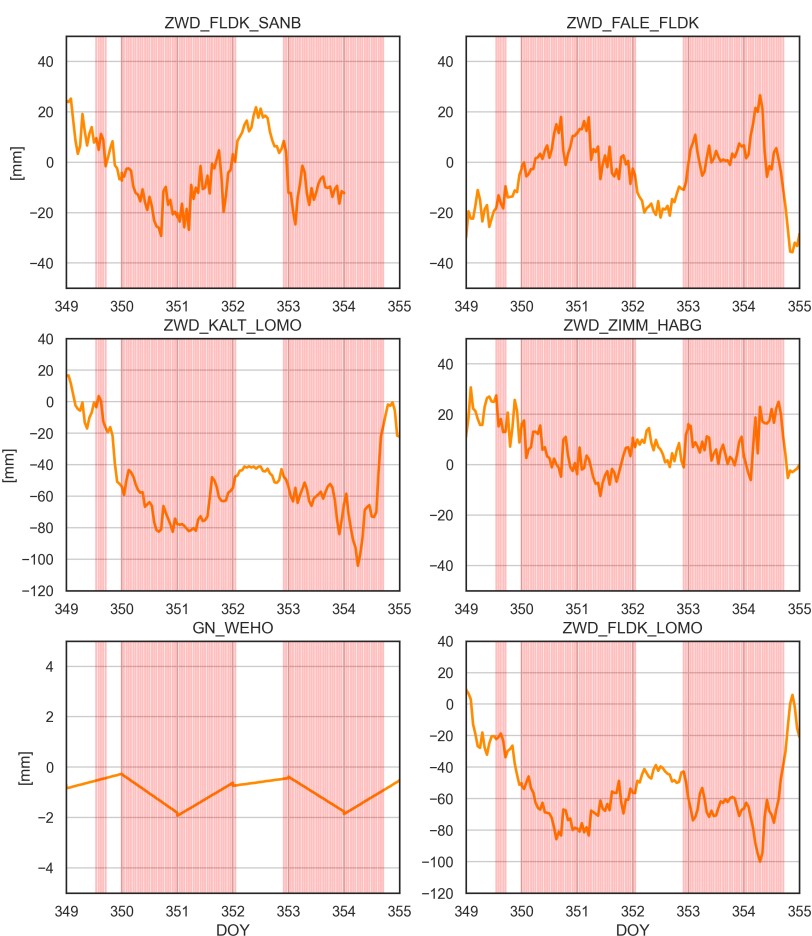

**Figure 8.** Time series of the six top predictors of the GB algorithm (most important features as shown in Figure 5) for the analysed time period (15.-21.12.2019). Observed foehn events (FI = 1) are color-coded in red.

## 6 Conclusions and outlook

In the present study, we introduced a new method for the detection and prediction of foehn events at the Swiss station Altdorf based on GNSS troposphere products and ML-based classification. We showed the performance of the introduced method by making use of an eleven-year-long data set of GNSS tropospheric parameters from a dedicated station network, namely the Swiss AGNES GNSS network as well as additional sites in neighbouring countries. Furthermore, we made use of FI observations at the SMN-station Altdorf for the chosen time period, provided by MeteoSwiss. In the course of an extensive cross-validation over the training data set (2010-2018), six different classification algorithms were tested. The two best-performing algorithms were the GB and SVC algorithms, which were then used in the following feature setup experiments. In a first experiment (FS1), we evaluated results of foehn classifications/predictions from those two algorithms over a two-year test period (2019-2020) at Altdorf. The second experiment (FS2) investigated the usability of NRT GNSS products for foehn prediction, in order to assess the feasibility of these low-latency ($\sim$ 30-40 minutes) data for operational forecasting. By comparing results from a third and fourth experiment (FS3 vs. FS4), we assessed the benefit of including single, geographically relevant, GNSS stations such as high-altitude sites. Additionally, the performance of the method was tested in a detailed investigation of a weekly period including two strong foehn events in December 2019.

The following main conclusions can be drawn from the presented results:

- The introduced ML-based method using GNSS troposphere products can provide encouraging results. It achieves equal performance in terms of both detection-based (POD = 75-80%, POFD = 1-2% ) and alarm-based (CAR = 66-76%, MAR = 1-2% ) metrics. On average, the results of both utilized algorithms are comparable to those obtained by Sprenger et al. (2017).

- The most promising results can be obtained if the full station network (shown in Figure 2) can be utilized. This incorporates also stations from neighbouring countries (Austria, Italy, Germany).

- When using NRT troposphere products instead of reprocessed data, e.g. for operational prediction, some degradation of the results has to be accepted. This shows differently for the two used algorithms. For SVC, the performance loss mainly concerns alarm-based measures whereas for the GB algorithm both performance metric types are affected similarly. There are two apparent reasons for the experienced degradation related to the GNSS products. First, the NRT products do not include tropospheric gradients which show up in the top predictors for the other case studies, at least for dedicated stations. Second, it can be expected that the quality of the prediction results also is influenced by the quality of the troposphere products. For NRT products, this quality is lower due to lower-quality (but also lower latency) products used in GNSS processing (such as satellite orbits and clocks). In order to confirm this assumption, we plan to compare results using the NRT setup (using only delay-products) on both NRT and reprocessed products in a possible future study.

- The final decision on which algorithm (GB or SVC) to use is left to the actual users. Both algorithms show equal performance for combined measures (COMB and $F_2$), with GB providing a more balanced performance in terms of detection-

and alarm-based statistics. Another major advantage of the GB algorithm is the availability of feature importance information, which can provide valuable insight in the physical relations and meaningfulness of the results. Furthermore, its performance for the detailed event analysis presented in Section 5 was clearly better (significantly lower number of false alarms).

– As expected from the physical background of GNSS troposphere products, ZWD differences from geographically important stations serve as the most important predictors. Still, also other chosen parameter types such as north and east gradients as well as ZHD differences (representing pressure gradients) show up in the list of top features for all experiments. Furthermore, it is worth to note that not only stations in the vicinity of Altdorf, but also further away (e.g. the Valais area or the Rhine valley) play an important role. Therefore also stations from neighbouring countries can be crucial for a good performance (such as station Feldkirch (FLDK), Austria). Using the special setup of FS3, the benefits of having additional data from high-altitude stations (SANB, OALP, ZERM) was outlined. Having the positive impacts of these stations on the result could also allow for shortening the amount of data for training (from nine to four years in this case), but lost foehn events due to missing data at added stations have to be kept in mind.

– Choosing the optimal performance metrics and appropriate pre-processing is a key task in ML-based classification algorithms, especially when working with such a highly imbalanced data set as in this study. The actual choice for the most important metric(s) strongly depends on the actual application of the prediction method, deciding whether detection- or alarm-based measures should be preferred. Within this study we tried to tune the algorithms for an optimal balance between both metric-types and leave a possible decision to the potential users. However, as already outlined before, there exists a trade-off between POD and CAR and therefore an optimization towards one metric will always result in a shortcoming towards the other.

Overall, these initial results are promising, and the developed method might aid the meteorological community as an additional tool for foehn detection and/or prediction as soon as the current major limitations can be mitigated. These limitations of the introduced method include:

– The most apparent limitation is that the availability of results from our method directly relies on the availability of GNSS data from all incorporated stations. As soon as data from only one station of the training network is missing, no results can be provided. This leads to a significant amount of periods (up to half of events in some cases) for which no results can be produced, as seen e.g. in Section 5.

– Station specificity: This study only shows the applicability of the method for the FI station Altdorf. The performance achieved at this location cannot be generalized for other stations or a whole region. Tests for other meteorological stations that regularly experience foehn, both in Switzerland and neighbouring countries, are planned for future studies (see possible improvements below).

– Supervised learning/dependence on target observations: As we use supervised learning techniques in this study, the results produced by our method will always be worse in comparison to the FI provided by MeteoSwiss, as this serves

as our target observation. However, this might change when both GNSS-based FI and operational FI are compared to another, independent reference data set (such as human forecasts).

– Detection-based measures are not yet at the level of studies with meteorological data. However, alarm-based measures were higher for our study, so a combination of data sets is expected to be beneficial for the overall performance.

– Looking into specific foehn events, the introduced method shows weaknesses in terms of capturing the exact periods of foehn onset and decay, compared to the standard FI algorithm. As only a one-week period (including two major foehn events) was analysed here, this result cannot be generalized yet. Therefore, further studies are needed to look into more

events in detail and, if needed, develop ways to increase the accuracy of the methods in terms of event start and end prediction.

Some of these limitations might be overcome by enhancements which can still be achieved through more detailed investigations in future studies. Possible improvements of the method we aim to investigate in the future would be:

– As stations from neighbouring countries are found to be important for the performance of the methods, extending the

550 utilized station network for several more of them is expected to benefit the results. Especially having a denser station distribution on the southern side of the Alps (Italy) should have a positive impact on the performance of the methods.

– Introduce time lags of specific predictors, accounting for the response time of the actual parameter at a specific station to changes in the synoptic conditions indicating a foehn event or not. Pressure-related parameters (ZHD) might be applied a greater time lag (response in advance), compared to humidity/temperature-related parameters (such as ZWD). As same

relation holds for the geographic location of contributing stations (south stations should response in advance). Such approaches might help to overcome the limitations in terms of exact onset/decay prediction.

– Use the developed method at other foehn locations in Switzerland and neighbouring countries. For some of those locations, the currently used setup might even provide better results than for Altdorf, based on a denser station network in those areas (e.g. Valais area).

– Enhance the nowcasting capabilities of the proposed method by including GNSS atmosphere gradients in the NRT products. If possible, gradient parameters should also be estimated at the same rate as delay products (every hour). Currently, this is done only every 6-12 hours and hourly estimates in between are interpolated linearly. This adaption might make the detection of smaller foehn events (lasting only a few hours) easier or even possible.

– Mitigation of data gaps in GNSS time series: Actually a number of different strategies can be investigated such as:

– Interpolation techniques: should be explored at least for shorter gaps.

– Training a set of different algorithms, iteratively excluding GNSS data from one station in the feature setup each time. This could provide a continuous solution under the assumption that only data from one station is missing.

- Using alternate features in the case of a particular feature missing, which is possible for ML algorithms relying on weak learners.

- Optimization of the methods' performance by carrying out a more extensive grid search for hyperparameter tuning of the used algorithms or try new (possibly more sophisticated deep learning) algorithms.

- Test the incorporation of GNSS products (especially from stations showing large impact in this study already) into algorithms based on meteorological data (as Sprenger et al. (2017)). As mentioned before, this might allow for a performance increase, especially for alarm-based statistics.

- Comparison to independent reference data: As mentioned in the list of limitations, the GNSS-based FI will always perform worse than the opertional FI in direct comparison. Therefore, it would be of interest to compare GNSS-based results with other, independent methods such as forecasts by human experts.

*Code and data availability.* Source code, trained algorithms and test data sets are available on request from the corresponding author.

## Appendix A: List of GNSS stations

This section provides the full GNSS station network which is utilized for this study.

**Table A1.** Full list of GNSS stations utilized for this study. Indicated are Name, geographical location (Longitude, Latitude and Height), Network to which a station belongs to as well as the Feature setup in which it is used.

| Name | Longitude [°] | Latitude [°] | Height [masl] | Network | Feature setups |
|------|---------------|--------------|---------------|---------|----------------|
| AIGE | 6.128259 | 46.247775 | 473.8768 | AGNES | All |
| BOU2 | 7.230437 | 47.394055 | 941.9955 | AGNES | All |
| DAV2 | 9.843516 | 46.812917 | 1645.5747 | AGNES | All |
| EPFL | 6.567896 | 46.521467 | 460.4702 | AGNES | All |
| ETHZ | 8.510532 | 47.407070 | 594.8398 | AGNES | All |
| FALE | 9.230295 | 46.804491 | 1344.1583 | AGNES | All |
| FLDK | 9.580601 | 47.231347 | 570.3447 | BEV | All |
| HABG | 8.182777 | 46.747459 | 1147.8459 | AGNES | All |
| HOH2 | 7.762704 | 46.319408 | 985.7388 | AGNES | All |
| HOHT | 7.762704 | 46.319408 | 985.7388 | AGNES | All |
| HUTT | 7.834883 | 47.141075 | 779.1001 | AGNES | All |
| KALT | 9.008414 | 47.217961 | 477.0371 | AGNES | All |
| KOPS | 10.115432 | 46.973951 | 1906.5541 | BEV | All |
| KREU | 9.160039 | 47.641294 | 529.9748 | AGNES | All |
| LECH | 10.139078 | 47.224056 | 1822.8005 | BEV | All |
| LOMO | 8.787428 | 46.172565 | 437.9931 | AGNES | All |
| LUZE | 8.300642 | 47.068204 | 542.2217 | AGNES | All |
| MAR2 | 7.070694 | 46.122154 | 644.1085 | AGNES | All |
| NEUC | 6.940483 | 46.993829 | 504.6724 | AGNES | All |
| OALP | 8.673724 | 46.660062 | 2139.4693 | AGNES | FS3 |
| PAYE | 6.943941 | 46.812141 | 548.7010 | AGNES | All |
| PFA2 | 9.784663 | 47.515328 | 1090.0990 | EUREF | All |
| SAAN | 7.301290 | 46.515572 | 1419.5502 | AGNES | All |
| SANB | 9.184548 | 46.463831 | 1702.2351 | AGNES | FS3 |
| SCHA | 8.655846 | 47.737566 | 638.1986 | AGNES | All |
| SIGM | 9.223912 | 48.083589 | 645.2889 | SAPOS | All |
| STA2 | 8.941636 | 45.855855 | 417.2265 | AGNES | All |
| STCX | 6.501172 | 46.822386 | 1155.4289 | AGNES | FS3 |
| STGA | 9.345949 | 47.441769 | 753.7296 | AGNES | All |
| TORI | 7.661280 | 45.063367 | 310.7408 | EUREF | All |
| WEHO | 7.472834 | 46.382054 | 2966.9314 | COGEAR | All |
| ZERM | 7.731996 | 46.001444 | 1931.1722 | AGNES | FS3 |
| ZIMM | 7.465275 | 46.877097 | 956.3256 | AGNES/EUREF/IGS | All |

*Author contributions.* MAR developed the research idea, software implementation and prepared the manuscript (including formal analysis, visualization, and writing). EB produced and provided the long-term time series of GNSS troposphere products used for this research. LC prepared selected visualizations for the manuscript. All co-authors contributed to the discussion of results as well as reviewing and editing the manuscript.

*Competing interests.* The authors declare that no competing interests are present.

*Acknowledgements.* The authors would like to thank swisstopo for providing GNSS troposphere products from the AGNES network and MeteoSwiss for providing foehn index and other meteorological observations for the station Altdorf. Furthermore, we thank the three anonymous reviewers for their valuable comments and suggestions which greatly helped to improve the manuscript.

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
