# Peer review of "Machine learning-based prediction of Alpine Foehn events using GNSS troposphere products: First results for Altdorf, Switzerland"

_Atmospheric Measurement Techniques, 2022_

## Referee Comment (RC1)

February 17, 2022

**Prediction of Alpine Foehn from time series of GNSS troposphere products using machine learning**

Aichinger-Rosenberger et al. *amt-2022-33*

**A. General Comments**

With foehn winds, air descends behind a topographic obstacle and often lowers moisture content there. The paper attempts to exploit this characteristic to diagnose and nowcast the occurrence of foehn (the response variable) with GPS-satellite derived integral moisture content variables as covariates. The response is an independently derived foehn indicator at a single station in Switzerland and the covariates are integral moisture measurements, their spatial gradients and horizontal differences among a few dozen stations in Switzerland. This is a unique, not yet exploited data set for the diagnosis and nowcasting of foehn. It could therefore be used to gain a better understanding of the mutual effects of foehn and moisture fields and to better diagnose or nowcast the occurrence of foehn but the paper falls short in both aspects as described in section B.

The results of the manuscript in its current form cannot be reproduced. Not enough details are given of the specific settings for the various machine learning algorithms and the data sets are also not available. Worryingly, a substantial amount of observed foehn events in Figs. 5, 6, 7, 9, 10 are absent, which casts doubt also on proper data handling for the rest of the paper.

**B. Specific Comments**

B.1 Improper data handling: The almost complete absence of foehn events in the last quarter of 2020 (which is part of verification period in the paper) from Figs. 5, 6, 7, 9, 10 seems suspicious since fall is a main foehn season. Indeed, a retrieval of the foehn index data for Altdorf for the period Sept-Dec 2020 shows 599 10-minute intervals with foehn instead of what seems to be only a single data point in the paper. Such a mistake casts serious doubt on proper data handling in the rest of the paper; especially since no summary statistics of the data are given, e.g. percentage of missing values in both response and covariates, range of values for the covariates. The same data check for Sept-Dec 2020 revealed that 240 data points of the foehn index are missing. Are those missing response dates properly excluded from the computation of the scores?

B.2 Not reproducible: Since neither code (only upon request) nor data are available and almost no specifics about the settings of the machine learning algorithms nor the version of the software package are given the results cannot be reproduced; even a plausibility check for appropriateness of the algorithm settings is not possible. And: how many covariates are actually used (add to table 1)?

B.3 Choice of machine learning algorithms: Why are exactly the algorithms listed in subsection 4.1 chosen among many possible candidates and why are so many used (see next issue)? Since random forests as ensembles of decision trees outperform them: why are decision trees included? Support vector classifiers assume a linear boundary between two classes (foehn/no foehn in this case) whereas support vector *machines* can handle non-linear boundaries. Why are SV classifiers chosen instead of SV machines?

B.4 Lack of performance optimization: If best possible performance of foehn diagnosis/nowcasting with GPS data is a goal then the setting of all algorithms should be tuned first instead of using default settings in the particular software package to select two and only tune these two. Other methods - if properly tuned - might work better.

B.5 Lack of physical understanding: The application of integral water vapor information from GPS satellites to the diagnosis of foehn is unique. Therefore an attempt is needed to understand details of the integral water vapor fields and their relation to foehn. Since most information lies in the ZWD field (cf. Fig. 8) figures with its average spatial distribution during foehn events and non-foehn events (of similar sample size as foehn events) will be helpful – similar to Sprenger et al. (2017) for pressure. Such maps should ideally be stratified by season. Since water vapor content is highly variable, an exploration behind the reasons of success and failure of the model diagnosis should be undertaken. Deep foehn situations, for example, might have a strong humidity gradient across the Alps, whereas shallow foehn cases or the onset of foehn events might have weak gradients. Since the GNSS stations are not collocated with the foehn station, consequences for the model performance should be explored, e.g. with maps of ZWD for foehn situations. A different avenue to pursue for increasing understanding is using an individual tree from a random forest model to illustrate how that model separates foehn cases from no-foehn cases.

B.6 Ultimate reason for the method: Why should foehn be diagnosed from GNSS-derived information? Weather station data give a more reliable answer for specific locations and such information was actually used as truth to approximate with GNSS data and machine learning algorithms. The method described in the paper cannot be used to diagnose foehn in locations without weather stations either, since it was trained on only one station and the transferability to other locations is not shown in the paper. The paper uses the approach for nowcasting 1 hour into the future and mentions that NWP models fare poorly with foehn quoting a paper from 2012. NWP models and their spatial resolution have dramatically improved in the decade since then. I would guess that MeteoSwiss has a current performance evaluation of COSMO1 available for Altdorf, against which the results of the paper could be measured. Results should also be compared to a simple persistence model, i.e. nowcasting the same no/foehn state as in the current hour.

B.7 Larger data set: To become more confident about the usability of integral moisture data for foehn diagnosis, more foehn locations should be included. Several more locations exist in Switzerland, for which a foehn index is available. To get more robust error estimates and performance scores, using the longer data set 1999-2020 mentioned in line 120 would be helpful. Line 125 merely states that only 2010-2020 is used without giving a reason.

B.8 Verification: Comparing total number of foehn hours from foehn index and the algorithms - stratified by season - should give an overall impression of the performance. To get an impression of the performance, a week-long time series containing one or more foehn events should be shown that includes the foehn index and the values of the four dominant features (as given in Fig. 8); if possible together with meteorological data of wind speed and direction, relative humidity and temperature.

Less crucial items to be changed are:

B.1 Give a short summary of how hydrometeors affect ZWD and ZHD and what that means for the applicability of the data set to foehn diagnosis, since foehn can happen with and without precipitation-sized particles.

B.2 Performance metrics: Subsection 4.4. can be shortened drastically by giving a confusion matrix and listing the scores derived from it in a table. After all, these are well-known scores in literature. Numbers for the confusion matrix should be given for both the test and training period.

B.3 Performance might be improved further by having GNSS information further south. Are there no such stations in Italy?

B.4 Focus the machine learning aspects in the introduction only on classification, the task at hand in the paper. You might add a further machine learning method to foehn diagnosis, namely mixture models (Plavcan et al., 2014).

B.5 Why are 12-hour moving average values of ZWD used in Fig. 2? Is the averaging window centered or asymmetric? Are the covariates used in the algorithms also 12-hour moving averages or "hourly troposphere products" as line 265 states? Does "hourly" mean an average over the hour or an instantaneous value every hour?

**C. Technical Corrections**

C.1 Fig. 1: Add topography and draw the lines between the stations, which contribute the top 4 features shown in Fig. 8.

C.2 Combine figures and tables to ease comparison: table 4 with table 5; Fig. 9 with Fig. 10 and rearrange Fig. 5 to have "observed" in center and the results from both methods immediately above and below (also bring lines closer together - a small detail that helps comparison).

**References**

Plavcan, David, Georg J. Mayr, and Achim Zeileis (2014). "Automatic and Probabilistic Foehn Diagnosis with a Statistical Mixture Model". In: *Journal of Applied Meteorology and Climatology* 53, pp. 652–659. DOI: 10.1175/jamc-d-13-0267.1.

Sprenger, Michael et al. (2017). "Nowcasting Foehn Wind Events Using the AdaBoost Machine Learning Algorithm". In: *Weather and Forecasting* 32.3, pp. 1079–1099. DOI: 10.1175/waf-d-16-0208.1.

---

## Author Comment (AC1)

**Answers to Comments Referee 1**

**General response**

First of all, we want to thank referee one for this extensive review and a lot of valuable comments and suggestions on how to improve the manuscript. In the following, we will go through all specific comments and give some first suggestions on strategies how we will implement the referees' points to improve the manuscript.

**Specific comments**

1. *Improper data handling: The almost complete absence of foehn events in the last quarter of 2020 (which is part of verification period in the paper) from Figs. 5, 6, 7, 9, 10 seems suspicious since fall is a main foehn season. Indeed, a retrieval of the foehn index data for Altdorf for the period Sept-Dec 2020 shows 599 10-minute intervals with foehn instead of what seems to be only a single data point in the paper. Such a mistake casts serious doubt on proper data handling in the rest of the paper; especially since no summary statistics of the data are given, e.g. percentage of missing values in both response and covariates, range of values for the covariates. The same data check for Sept-Dec 2020 revealed that 240 data points of the foehn index are missing. Are those missing response dates properly excluded from the computation of the scores?*

    Thanks for cross-checking the foehn event data, this is a very valuable and the most obvious point for improvement, which we already have addressed. After a first check of the corresponding code, we encountered a bug when mapping the 10-min foehn index observations to full hours. Furthermore, missing GNSS products at certain times have also caused the exclusion of certain timestamps of the training/testing periods. We have already improved the corresponding code and will include detailed statistics on the overall number of foehn events as well as events which had to be excluded due to lack of (GNSS) training data. The missing response dates are also excluded. We will provide extensive statistics of available/useable data (both GNSS and Foehn index) in the revised manuscript.

2. *Not reproducible: Since neither code (only upon request) nor data are available and almost no specifics about the settings of the machine learning algorithms nor the version of the software package are given the results cannot be reproduced; even a plausibility check for appropriateness of the algorithm settings is not possible. And: how many covariates are actually used (add to table 1)?*

    The code and data set (pre-processed data, i.e. feature matrix for the test cases) will be made available with the revised manuscript in form of an online repository. We will also try to give more details on the algorithms used in the revised version of the manuscript. The software package is the latest version of sklearn (1.0.2). The number of covariates will be added.
    However, we would also like to mention that comparable studies (such as Sprenger et. al (2017)) also did not publish any additional material (neither code nor data), therefore we still see the upload of additional material as optional.

3.  *Choice of machine learning algorithms: Why are exactly the algorithms listed in subsection 4.1 chosen among many possible candidates and why are so many used (see next issue)? Since random forests as ensembles of decision trees outperform them: why are decision trees included? Support vector classifiers assume a linear boundary between two classes (foehn/no foehn in this case) whereas support vector machines can handle non-linear boundaries. Why are SV classifiers chosen instead of SV machines?*

The original idea was to test several algorithms of different types and find out which ones work best for this task. Those algorithms (which are tested) were also described in the introduction, indicating that these have already been used (with success) in similar studies. Therefore, we chose them for the cross-validations. This reason will be added in the manuscript for clarification. For sure there are a lot more possibilities to choose from but at some point, a decision for one/several algorithms had to be made. However, it is reasonable to exclude decision trees here, which we will do for the revised manuscript.
Concerning Support Vector Classifiers/Machines: The implementation of SVC in sklearn can also handle non-linear boundaries. Therefore, it takes the keyword "kernel" which can be set to e.g. "polynomial" but also "linear" in case one wants linear boundaries. The default value would be RBF (Radial Basis Function), which was set in our analyses. We will include information on these settings in the manuscript.

4.  *Lack of performance optimization: If best possible performance of foehn diagnosis/nowcasting with GPS data is a goal then the setting of all algorithms should be tuned first instead of using default settings in the particular software package to select two and only tune these two. Other methods - if properly tuned - might work better.*

The main goal of the study is not finding the ultimate best performing algorithm for usage of GNSS data to detect/predict foehn events but rather showing that this is possible at all using machine-learning based classification (proof-of-concept). Although we already do quite a bit of fine tuning for the two selected algorithms, applying an extensive grid-search for one specific algorithm would be an interesting investigation of a follow-up study. We realized that therefore we need to adjust some formulations in the manuscript (such as the title even), which might be too promising for potential readers.
As mentioned in the last point, we wanted to start out this initial study with a cross-comparison of several algorithms to find potential candidates for further optimization for dedicated case studies. Therefore, we chose the default settings of all algorithms (i.e. their sklearn implementation) in the cross-comparison as this seems to secure the most independent assessment possible. We are still convinced that this a reasonable way to approach this problem, since no prior experience on which algorithm (and which settings) will work best is available (when using GNSS data for training). Thus, using default settings (which work best for a majority of classification problems) should be the most objective choice in our opinion. For sure some (not chosen) algorithms might outperform the ones proposed here after extensive tuning. However, at some point one will also face time constraints as this potentially becomes a (possibly) never-ending optimization of settings.
For these reasons, we want to stick to our initial approach, however we will still fine-tune one or two other algorithms (e.g. Random Forest) and evaluate their performance for the case studies as well.

*5. Lack of physical understanding: The application of integral water vapor information from GPS satellites to the diagnosis of foehn is unique. Therefore, an attempt is needed to understand details of the integral water vapor fields and their relation to foehn. Since most information lies in the ZWD field (cf. Fig. 8) figures with its average spatial distribution during foehn events and non-foehn events (of similar sample size as foehn events) will be helpful – similar to Sprenger et al. (2017) for pressure. Such maps should ideally be stratified by season. Since water vapor content is highly variable, an exploration behind the reasons of success and failure of the model diagnosis should be undertaken. Deep foehn situations, for example, might have a strong humidity gradient across the Alps, whereas shallow foehn cases or the onset of foehn events might have weak gradients. Since the GNSS stations are not collocated with the foehn station, consequences for the model performance should be explored, e.g. with maps of ZWD for foehn situations. A different avenue to pursue for increasing understanding is using an individual tree from a random forest model to illustrate how that model separates foehn cases from no-foehn cases.*

This is certainly a very valuable point, which will also be investigated further. However, we feel that we already have addressed the basic physical understanding through showing the response of ZWD values (and differences between them for station north/south of the Alpine ridge) in Figure 2. That is where the basic idea for this study comes from. Furthermore, Figure 8 already shows the importance of certain stations or differences between them (such as HABG and SANB or KALT and SANB) and indicates that these differences have the highest impact on the classification. We will include additional plots of the differences in ZWD for these stations for selected foehn events which might show a physical connection and enhance physical understanding.
ZWD maps: Although we have some means to do so at hand, ZWD maps are not trivial to produce. Therefore, we are not sure if we already can incorporate such maps in this study or try to address this point in a follow-up study.

*6. Ultimate reason for the method: Why should foehn be diagnosed from GNSS-derived information? Weather station data give a more reliable answer for specific locations and such information was actually used as truth to approximate with GNSS data and machine learning algorithms. The method described in the paper cannot be used to diagnose foehn in locations without weather stations either, since it was trained on only one station and the transferability to other locations is not shown in the paper. The paper uses the approach for nowcasting 1 hour into the future and mentions that NWP models fare poorly with foehn quoting a paper from 2012. NWP models and their spatial resolution have dramatically improved in the decade since then. I would guess that MeteoSwiss has a current performance evaluation of COSMO1 available for Altdorf, against which the results of the paper could be measured. Results should also be compared to a simple persistence model, i.e. nowcasting the same no/foehn state as in the current hour.*

As already mentioned before, the aim of this study was not to give an ultimate, stand-alone method for foehn detection/prediction but rather showing that the method introduced denotes an additional tool that might be helpful to achieve those tasks. The main aim is still a proof-of-concept for a completely new approach. Therefore, we will scale back some of the promises given in the manuscript (e.g. in the title). Also, this study (just as Sprenger et. al (2017)) focuses on detecting/predicting foehn events at the specific location of Altdorf. Therefore, we will add this information to the

title. In a further study we plan to extend the investigations to other foehn locations (in Switzerland or also Austria), which for sure will require dedicated training of the algorithms at the respective stations. Therefore, the method will always be somehow specific for a certain location. Concerning NWP performance and evaluation, we will contact MeteoSwiss for some details on the most recent performance.

7. *Larger data set: To become more confident about the usability of integral moisture data for foehn diagnosis, more foehn locations should be included. Several more locations exist in Switzerland, for which a foehn index is available. To get more robust error estimates and performance scores, using the longer data set 1999-2020 mentioned in line 120 would be helpful. Line 125 merely states that only 2010-2020 is used without giving a reason.*

It was the original plan to use all available data (1999-2020) for this study. However, a number of stations which are important for the method (as seen in the feature importance plot) are not available since 1999. Therefore, one will lose those products/stations as features/predictors for the entire classification as all used features have to be available for the whole training and test period. This was the ultimate reason to choose the timespan 2010-2020. We will add this information in the manuscript. Extension of the approach to other (foehn index) stations is planned for a follow-up study.

8. *Verification: Comparing total number of foehn hours from foehn index and the algorithms- stratified by season - should give an overall impression of the performance. To get an impression of the performance, a week-long time series containing one or more foehn events should be shown that includes the foehn index and the values of the four dominant features (as given in Fig. 8); if possible together with meteorological data of wind speed and direction, relative humidity and temperature.*

This comparison is actually already done/included when computing the performance measures but including overall values as a separate statistic is also possible. Stratification by season is a good idea which will be implemented by us. As suggested, we will also add such a week-long time series plot, also including meteorological data as this will make it easier to see the performance for dedicated events and increase physical understanding.

**Less crucial points**

1. *Give a short summary of how hydrometeors affect ZWD and ZHD and what that means for the applicability of the data set to foehn diagnosis, since foehn can happen with and without precipitation-sized particles*

This is a topic where only little research has been done so far, but we will incorporate 2-3 relevant studies in the introduction.

2. *Performance metrics: Subsection 4.4. can be shortened drastically by giving a confusion matrix and listing the scores derived from it in a table. After all, these are well-known scores in literature. Numbers for the confusion matrix should be given for both the test and training period*

Performance metrics: we will shorten this section as suggested and give the confusion matrix.

3. *Performance might be improved further by having GNSS information further south. Are there no such stations in Italy?*

   Yes, there are stations in Italy, but we don't have data available (yet). The inclusion of more stations from different regions is also planned for a follow-up study.

4. *Focus the machine learning aspects in the introduction only on classification, the task at hand in the paper. You might add a further machine learning method to foehn diagnosis, namely mixture models (Plavcan et al., 2014).*

   This point will be addressed in the revised manuscript.

5. *Why are 12-hour moving average values of ZWD used in Fig. 2? Is the averaging window centered or asymmetric? Are the covariates used in the algorithms also 12-hour moving averages or "hourly troposphere products" as line 265 states? Does "hourly" mean an average over the hour or an instantaneous value every hour?*

   Moving-average filtering is applied to reduce noise present in the ZWD estimates. The moving average is centered and there is no particular reason for choosing a 12-hour window beside optimized performance in noise reduction (for visualization). The ZWD estimates used were originally also filtered, which also explains "lost" foehn events at the very end of the test period (last hours of 31.12.2020). However, recent tests reveal comparable performance using unfiltered estimates. Therefore, we will present results using unfiltered ZWD data in the revised manuscript, in order to not lose additional foehn events.
   The "hourly" GNSS products are estimated every hour from 30-second measurements together with station coordinates (in this case using least-squares adjustment). We will add this information to the manuscript.

**Technical corrections**
These will be included in the revised version.

---

## Author Comment (AC2)

**Answers to Comments Referee 2**

**General response**

*The paper „Prediction of Alpine Foehn from time series of GNSS troposphere products using machine learning" shows first the selection of the ML methods and then usage of two of them on a GNSS tropospheric data set (tropospheric delays and gradients) to detect the foehn occurrences. It is a very new field of study as most of the GNSS meteorology research focuses on the precipitation/humidity parameters rather as foehn. Also the usage of the machine learning algorithms is interesting. I found the paper very well written. The only drawbacks of the paper are: 1. Sometimes a more extended discussion on the results is lacking, 2. The figures (especially Fig.5-10) could be made more interesting.*

We want to thank referee two for the positive feedback on the manuscript and valuable comments how to further improve it. We will provide a more detailed discussion of certain aspects and try to make the plots more interesting (as far as possible). All specific comments are addressed below.

**Specific comments**

1. *Title: since you always work on the past data (even with the NRT approach), it is rather a 'detection' than a 'prediction', so maybe the title could be changed accordingly*

   We were already thinking about changing the title (e.g. specifying the location Altdorf in the title) as the original title might be too promising. However, as we introduce the time shift on the FI time series, we actually do a prediction (for the next hour). We might provide some results for both approaches (detection—shift = 0 h and prediction—shift = -1 h) for comparison.

2. *Line 3: 'lee/luv' – a specific terminology, maybe worth explaining (at least in the Introduction, however 'luv' doesn't appear anywhere else than the abstract*

   Will be introduced

3. *Line 68 'COSMO (Consortium for Small-scale Modeling).' -> 'Consortium for Small-scale Modeling COSMO)'; the full name should go before abbreviation*

   Will be changed

4. *Line 90: This is not the exact formula from Rueger and I think there is a mistake there: However, I would recommend sticking to the original formulation as then you have a clear distinction between the dry and water vapor parts.*

   Thanks for this hint, we will include the original version in the revised manuscript.

5. *Figure 1: Would be nice to see the topography in this Figure to better visualize foehn*

   Will be updated

6. *Section 4.1: I would recommend giving here at least very brief overview of the selected methods*

Will be updated

7. *Line 163: '(negative) maximum' - > why not use 'minimum' here?*

We would interpret 'minimum' as close to zero but one can for sure argue to use minimum here as well.

8. *Fig.3 and Table 2 show exactly the same information, so I would recommend removing one of them, especially that Fig. 3 is not even addressed in the main text.*

This is true, we will remove Figure 3.

9. *Figure 4: Make the foehn line more pronounced*

Will be updated

10. *Line 259: Would be good to comment here what the chosen parameters mean*

Will be updated

11. *Figure 5: Maybe you could add vertical lines so the reader can more easily compare the data for particular dates; also you do not comment this plot in the text*

Will be updated and commented in the text

12. *Figure 6 and 7: Maybe there is a way to plot them together for better comparisons of the two methods?*

We will try to come up with a plot like this, although it is quite challenging to combine the plots while keeping the clarity

13. *Line 284: A more detailed discussion about the features would be advantageous*

Will be given

14. *Figure 9: Why not add here a line also of the match with GB (not only with the adjusted one); also it seems like the event of Oct 2020 was caught by the algorithm but in a different epoch – maybe it is something to look into*

Will be done, thanks for the hint

15. *Line 312: Would be nice to see here more discussion on why you change the threshold and how it is done*

Will be included in the revised manuscript

---

## Author Response (AR1)

**Answers to Comments Referee 1**

**General response**

First, we want to thank referee one for this extensive review and a lot of valuable comments and suggestions on how to improve the manuscript. In the following, we will go through all specific comments and give some details how we implemented most of the referees' points to improve the manuscript.

**Specific comments**

1. *Improper data handling: The almost complete absence of foehn events in the last quarter of 2020 (which is part of verification period in the paper) from Figs. 5, 6, 7, 9, 10 seems suspicious since fall is a main foehn season. Indeed, a retrieval of the foehn index data for Altdorf for the period Sept-Dec 2020 shows 599 10-minute intervals with foehn instead of what seems to be only a single data point in the paper. Such a mistake casts serious doubt on proper data handling in the rest of the paper; especially since no summary statistics of the data are given, e.g. percentage of missing values in both response and covariates, range of values for the covariates. The same data check for Sept-Dec 2020 revealed that 240 data points of the foehn index are missing. Are those missing response dates properly excluded from the computation of the scores?*

Thanks for cross-checking the foehn event data, this is a very valuable and the most obvious point for improvement, which we have addressed. We have improved the corresponding code and included detailed statistics on the overall number of foehn events as well as events which had to be excluded due to lack of GNSS data. Essentially, we have now defined criteria for the selection of stations and the study period to make sure we do not lose the majority of foehn events due to lack of GNSS data (one station missing is enough to not produce any result). The missing response dates are properly excluded.

2. *Not reproducible: Since neither code (only upon request) nor data are available and almost no specifics about the settings of the machine learning algorithms nor the version of the software package are given the results cannot be reproduced; even a plausibility check for appropriateness of the algorithm settings is not possible. And: how many covariates are actually used (add to table 1)?*

Code and test data set will be made available with the revised manuscript in form of an online repository, in order to make the results reproducible. We have also tried to give more details on the algorithms used in the revised version of the manuscript, although plenty thereof was

already given in section about hyperparamter tuning in the old version. The software package is the 1.0.1 version of sklearn. The number of covariates was added to the study setup tables.

However, we would also like to mention that comparable studies (such as Sprenger et. al (2017)) also did not publish any additional material (neither code nor data), therefore we still see the upload of additional material as optional.

3. **Choice of machine learning algorithms: Why are exactly the algorithms listed in subsection 4.1 chosen among many possible candidates and why are so many used (see next issue)? Since random forests as ensembles of decision trees outperform them: why are decision trees included? Support vector classifiers assume a linear boundary between two classes (foehn/no foehn in this case) whereas support vector machines can handle non-linear boundaries. Why are SV classifiers chosen instead of SV machines?**

The original idea was to test several algorithms of different types and find out which ones work best for this task. Those algorithms (which are tested) were also described in the introduction, indicating that these have already been used (with success) in similar studies. Therefore, we chose them for the cross-validation. For sure there are a lot more possibilities to choose from but at some point, a decision for one/several algorithms had to be made. However, it is reasonable to exclude decision trees here, which we did for the revised manuscript.

Concerning Support Vector Classifiers/Machines: The implementation of SVC in sklearn can also handle non-linear boundaries. Therefore, it takes the keyword "kernel" which can be set to e.g. "polynomial" but also "linear" in case one wants linear boundaries. The default value would be RBF (Radial Basis Function), which was set in our analyses.

4. **Lack of performance optimization: If best possible performance of foehn diagnosis/nowcasting with GPS data is a goal then the setting of all algorithms should be tuned first instead of using default settings in the particular software package to select two and only tune these two. Other methods - if properly tuned - might work better.**

The main goal of the study is not finding the ultimate best performing algorithm for usage of GNSS data to detect/predict foehn events but rather showing that this is possible at all using machine-learning based classification (proof-of-concept). Although we already do quite a bit of fine tuning for the two selected algorithms, applying an extensive grid-search for one specific algorithm would be an interesting investigation of a follow-up study. We realized that therefore we need to adjust some formulations in the manuscript (such as the title even), which might be too promising for potential readers.

As mentioned in the last point, we wanted to start out this initial study with a cross-comparison of several algorithms to find potential candidates for further optimization for dedicated case studies. Therefore, we chose the default settings of all algorithms (i.e. their sklearn implementation) in the cross-comparison as this seems to secure the most independent assessment possible. We are still convinced that this a reasonable way to approach this problem, since no prior experience on which algorithm (and which settings) will

work best is available (when using GNSS data for training). Thus, using default settings (which work best for a majority of classification problems) should be the most objective choice in our opinion. For sure some (not chosen) algorithms might outperform the ones proposed here after extensive tuning. However, at some point one will also face time constraints as this potentially becomes a (possibly) never-ending optimization of settings.

For these reasons, we have sticked to our initial approach but putting much more emphasis on the performance of the two chosen algorithms in the case studies.

5. ***Lack of physical understanding: The application of integral water vapor information from GPS satellites to the diagnosis of foehn is unique. Therefore, an attempt is needed to understand details of the integral water vapor fields and their relation to foehn. Since most information lies in the ZWD field (cf. Fig. 8) figures with its average spatial distribution during foehn events and non-foehn events (of similar sample size as foehn events) will be helpful – similar to Sprenger et al. (2017) for pressure. Such maps should ideally be stratified by season. Since water vapor content is highly variable, an exploration behind the reasons of success and failure of the model diagnosis should be undertaken. Deep foehn situations, for example, might have a strong humidity gradient across the Alps, whereas shallow foehn cases or the onset of foehn events might have weak gradients. Since the GNSS stations are not collocated with the foehn station, consequences for the model performance should be explored, e.g. with maps of ZWD for foehn situations. A different avenue to pursue for increasing understanding is using an individual tree from a random forest model to illustrate how that model separates foehn cases from no-foehn cases.***

This is certainly a very valuable point, which also was investigated further. However, we feel that we already have addressed the basic physical understanding through showing the response of ZWD values (and differences between them for station north/south of the Alpine ridge) in Figure 2 of the initial manuscript. That is where the basic idea for this study comes from. Furthermore, the new figures added which show the feature importance of the GB algorithm already show the importance of certain stations or differences between them (such as FLDK and SANB or KALT and LOMO) and indicates that these differences have the highest impact on the classification. We have included additional plots of the top predictors for selected foehn events which show the physical connection and enhance physical understanding.

ZWD maps: Although we have some means to do so at hand, ZWD maps are not trivial to produce.  Furthermore, the accuracy of such maps is limited by the density of the station network because of interpolation errors (especially with height). Switzerland might has one of the densest networks in Europe, but the horizontal resolution is still not comparable to a NWP grid or the SwissMetNet station network. Therefore, we did not include ZWD maps.

6. ***Ultimate reason for the method: Why should foehn be diagnosed from GNSS-derived information? Weather station data give a more reliable answer for specific locations and such information was actually used as truth to approximate with GNSS data and machine learning algorithms. The method***

*described in the paper cannot be used to diagnose foehn in locations without weather stations either, since it was trained on only one station and the transferability to other locations is not shown in the paper. The paper uses the approach for nowcasting 1 hour into the future and mentions that NWP models fare poorly with foehn quoting a paper from 2012. NWP models and their spatial resolution have dramatically improved in the decade since then. I would guess that MeteoSwiss has a current performance evaluation of COSMO1 available for Altdorf, against which the results of the paper could be measured. Results should also be compared to a simple persistence model, i.e. nowcasting the same no/foehn state as in the current hour.*

As already mentioned before, the aim of this study was not to give an ultimate, stand-alone method for foehn detection/prediction but rather showing that the method introduced denotes an additional tool that might be helpful to achieve those tasks. The main aim is still a proof-of-concept for a completely new approach. Therefore, we scaled back some of the promises given in the manuscript (e.g. in the title). Also, this study (just as Sprenger et. al (2017)) focuses on detecting/predicting foehn events at the specific location of Altdorf. Therefore, we have added this information to the title. In a further study we plan to extend the investigations to other foehn locations (in Switzerland or also Austria), which for sure will require dedicated training of the algorithms at the respective stations. Therefore, the method will always be somehow specific for a certain location. Concerning NWP performance and evaluation, we have not got any update on the performance and not found any new studies on it. Therefore, it seems reasonable to us to quote the latest one available, even when already from 2012, just as our reference study did.

7. *Larger data set: To become more confident about the usability of integral moisture data for foehn diagnosis, more foehn locations should be included. Several more locations exist in Switzerland, for which a foehn index is available. To get more robust error estimates and performance scores, using the longer data set 1999-2020 mentioned in line 120 would be helpful. Line 125 merely states that only 2010-2020 is used without giving a reason.*

It was the original plan to use all available data (1999-2020) for this study. However, several stations which are important for the method (as seen in the feature importance plot) are not available since 1999. Therefore, one will lose those products/stations as features/predictors for the entire classification as all used features have to be available for the whole training and test period. We have now defined dedicated criteria for the selection of GNSS stations and study periods, based on statistics about covered foehn events.

8. *Verification: Comparing total number of foehn hours from foehn index and the algorithms- stratified by season - should give an overall impression of the performance. To get an impression of the performance, a week-long time series containing one or more foehn events should be shown that includes the foehn index and the values of the four dominant features (as given in Fig. 8); if possible together with meteorological data of wind speed and direction, relative humidity and temperature.*

This comparison is already done/included when computing the performance measures but including overall values as a separate statistic is also possible. Specific stratification by season is not shown since for the two-year test period these statistics might not be robust, as the number of foehn events covered is already quite low in general. For a possible follow-up study (with a larger test data set) we might investigate the seasonal performance of the method. As suggested, we have also added a section showing such a week-long time series plot, also including meteorological data. Most of the top 6 predictors show expected behavior for the observed foehn periods. We hope this increases the physical understanding of the approach.

**Less crucial points**

1. *Give a short summary of how hydrometeors affect ZWD and ZHD and what that means for the applicability of the data set to foehn diagnosis, since foehn can happen with and without precipitation-sized particles*

We incorporated a section on this topic.

2. *Performance metrics: Subsection 4.4. can be shortened drastically by giving a confusion matrix and listing the scores derived from it in a table. After all, these are well-known scores in literature. Numbers for the confusion matrix should be given for both the test and training period*

Performance metrics: The section was shortened, and the confusion matrix is given for the case studies.

3. *Performance might be improved further by having GNSS information further south. Are there no such stations in Italy?*

A few stations from Italy are part of the available data set (as well as some Austrian and German stations), those are now part of the training and test data. For most other stations in Northern Italy, we don't have data available (yet). The inclusion of more stations from different regions is also planned for a follow-up study.

4. *Focus the machine learning aspects in the introduction only on classification, the task at hand in the paper. You might add a further machine learning method to foehn diagnosis, namely mixture models (Plavcan et al., 2014).*

This point has been addressed in the revised manuscript. *Plavcan et al., 2014* was added.

5. *Why are 12-hour moving average values of ZWD used in Fig. 2? Is the averaging window centered or asymmetric? Are the covariates used in the algorithms also 12-hour moving averages or "hourly troposphere products" as line 265 states? Does "hourly" mean an average over the hour or an instantaneous value every hour?*

Moving-average filtering is applied to reduce noise present in the ZWD estimates. The moving average is centered and there is no particular reason for choosing a 12-hour window beside optimized performance in noise reduction (for visualization). The ZWD estimates used were originally also filtered, which also explains "lost" foehn events at the very end of the test period (last hours of 31.12.2020). However, tests reveal comparable performance using unfiltered estimates. Therefore, we present results using unfiltered ZWD data in the revised manuscript, in order to not lose additional foehn events.

The "hourly" GNSS products are estimated every hour from 30-second measurements together with station coordinates (in this case using least-squares adjustment). We added this information to the manuscript.

**Technical corrections**

These have been included in the revised version.

**Answers to Comments Referee 2**

**General response**

*The paper „Prediction of Alpine Foehn from time series of GNSS troposphere products using machine learning" shows first the selection of the ML methods and then usage of two of them on a GNSS tropospheric data set (tropospheric delays and gradients) to detect the foehn occurrences. It is a very new field of study as most of the GNSS meteorology research focuses on the precipitation/humidity parameters rather as foehn. Also the usage of the machine learning algorithms is interesting. I found the paper very well written. The only drawbacks of the paper are: 1. Sometimes a more extended discussion on the results is lacking, 2. The figures (especially Fig.5-10) could be made more interesting.*

We want to thank referee two for the positive feedback on the manuscript and valuable comments how to further improve it. We have provided a more detailed discussion of certain aspects and removed/changed several plots. All specific comments are addressed below.

**Specific comments**

1. *Title: since you always work on the past data (even with the NRT approach), it is rather a 'detection' than a 'prediction', so maybe the title could be changed accordingly*

We have changed the title (e.g. specifying the location Altdorf in the title) as the original title might be too promising. However, as we introduce the time shift on the FI time series, we actually do a prediction (for the next hour).

2. *Line 3: 'lee/luv' – a specific terminology, maybe worth explaining (at least in the Introduction, however 'luv' doesn't appear anywhere else than the abstract*

Lee/luv terminology was left out of the entire manuscript

3. *Line 68 'COSMO (Consortium for Small-scale Modeling).' -> 'Consortium for Small-scale Modeling COSMO)'; the full name should go before abbreviation*

Has been changed

4. *Line 90: This is not the exact formula from Rueger and I think there is a mistake there: However, I would recommend sticking to the original formulation as then you have a clear distinction between the dry and water vapor parts.*

Thanks for this hint. Although we did not find a particularly different version of the formula, we have included a changed version (which shows the split in dry and wet part more clearly) in the revised manuscript. Maybe you could give the exact version you mean and we could still implement it in the manuscript.

**5. *Figure 1: Would be nice to see the topography in this Figure to better visualize foehn***

Has been updated

**6. *Section 4.1: I would recommend giving here at least very brief overview of the selected methods***

As the manuscript is quite long already, we actually would like to leave the reader here with the references given describing the methods.

**7. *Line 163: '(negative) maximum' - > why not use 'minimum' here?***

We would interpret 'minimum' as close to zero but one can for sure argue to use minimum here as well.

**8. *Fig.3 and Table 2 show exactly the same information, so I would recommend removing one of them, especially that Fig. 3 is not even addressed in the main text.***

Figure 3 (along with some others) has been removed.

**9. *Figure 4: Make the foehn line more pronounced***

Figure 4 was removed from the manuscript, as the cross-validation/algorithm selection was shortened

**10. *Line 259: Would be good to comment here what the chosen parameters mean***

Actually, this is done in the caption of (what is now) Table 3.

**11. *Figure 5: Maybe you could add vertical lines so the reader can more easily compare the data for particular dates; also you do not comment this plot in the text***

Figure 5 has been removed

**12. *Figure 6 and 7: Maybe there is a way to plot them together for better comparisons of the two methods?***

These plots have been removed, a direct comparison over one week can now be found in the last results section.

**13. *Line 284: A more detailed discussion about the features would be advantageous***

Is now included

**14. Figure 9: Why not add here a line also of the match with GB (not only with the adjusted one); also it seems like the event of Oct 2020 was caught by the algorithm but in a different epoch – maybe it is something to look into**

Figure 9 has also been removed

**15. Line 312: Would be nice to see here more discussion on why you change the threshold and how it is done**

The entire section on the threshold optimization has been removed from the manuscript, as the updated results suggest that it is not needed anymore (at least not as much for the old results)

**Author's changes in manuscript**

**Main changes:**

- **Title of the manuscript**:
  Title has been changed as the old one might have been to promising, reference to the station Altdorf was included
- **Detailed statistics of data availability**:
  Due to the extensive comments of Reviewer 1 on missing foehn events in the plots provided in the original manuscript, we now introduced detailed statistics on the data availability (for GNSS data mainly) for each case study. These missing periods originated in the fact that the GNSS time series contain data gaps for a number of stations. Since our method needs data from the full station network selected to be able to predict/detect at a certain time period, we cannot produce output if only one GNSS station is missing. This leads to a rather large number of foehn events for which we cannot produce predictions/detections. Detailed statistics on how many foehn events we are able to produce output per case study are now included in the manuscript. Furthermore, the newly chosen study setup is based on criteria influenced by the number of covered foehn events, see next point.
- **Selection criteria for station/feature setup**:
  We newly defined three selection criteria for the feature setup (mainly concerns the station selection) which are outlined in the new section …. Aim was to find a balance between using an appropriately dense GNSS network while still catching the majority of foehn events, in order to have a good amount of training data and more robust statistics.
- **Case studies:**
  The number of case studies was changed (from two to three), with the new one investigating the situation of having less data (only 5 instead of 10 years) but a better station distribution (additional high-altitude stations).
- **Section showing a detailed performance analysis for one week:**
  As requested by referee 1, we added this last section which shows two major foehn events in one week of December 2019, comparing our index against the operational one from MeteoSwiss. In addition, meteorological data (temperature, humidity and wind) as well as the six top predictors of our method are shown.
- **Plots in general:**
  The majority of the old plots were removed (except for feature importances) and only the tables with statistics were left in. As requested, confusion matrices are now plotted in addition.

**Some of the minor changes:**

- References on machine-learning based classification were shorten to exclusively atmospheric science studies (request from Referee 1)
- Mixture-models (Plavcan et. al, 2014) were added in the introduction
- Section on the influence of hydrometeors on GNSS troposphere estimates was added

- Equation for refractivity (Rueger paper) was changed (comment by Referee 2)
- Performance metrics section was shortened
- New station map with the newest station setup (and setups distinguished between case studies) as well as topography was added

---

## Referee Report (RR1)

**Machine-learning based prediction of Alpine Foehn events using GNSS troposphere products: First results for Altdorf, Switzerland**

**Aichinger-Rosenberger et al. *amt-2022-33rev1**

**A. General Comments**

The incorporation of reviewers' comments into the revised manuscript has improved it considerably. The title change already signals the intention of the paper to prove a concept – how to use previously untapped measurements for the diagnosis and nowcasting of foehn.

A few sticky issues remain or have surfaced in the course of the major rewrite of the article. They are listed in the next section.

Some serious mistakes in the computations had been found in the original manuscript. They seem to not have been completely eliminated and top of it, some of the figures that would have helped in cross-checking the results have been dropped in the revision. The first item of the "Specific Comments" gives details.

**B. Specific Comments**

B.1 **Correctness of some computations in doubt**: The review of the original manuscript version had unearthed large numbers of missed events in the observations of the foehn index FI (even though they are not missing at the MeteoSwiss database as I could confirm; original Fig. 5) and the events diagnosed from GNSS measurements (original Figs. 5–7 and Figs. 9-10). How the authors "have improved the corresponding code" remains vague in the response. I would have expected to see at least one figure showing observed and diagnosed events in the revised version but these figures have been completely dropped, which eliminates a chance to at least visually inspect the appropriateness of the results. That raises nagging doubts .... What is left for a cross-check are the performance metrics. If coding mistakes had been made only in the way missing GNSS data are handled (as the response indicates), differences should be seen in the performance metrics of the test data *and* the cross-validated training data. However, the numbers for the cross-validated training data metrics in the original and the revised version (Table 2 in both) are identical whereas the metrics for the test data have changed. As far as I see, some computations still have to be wrong. Consequently, the manuscript should not be published unless the correctness of the computations can be convincingly shown.

B.2 **Selection and application of machine-learning methods**: It is important to point out in the paper that no universal truth exists for the diagnosis of foehn and even diagnoses by human experts vary considerably (Mayr et al., 2018). Therefore the classification problem belongs to the category of unsupervised learning (e.g. Hastie et al., 2009). The paper, on the other hand, uses supervised learning methods, a choice that needs justification. A consequence of the use of supervised methods is raised in the following issue.

B.3 **Dependence on foehn classified with traditional meteorological measurements**: Despite the claim of the paper (line 445 and lines 521–522) the foehn classification with GNSS data is completely dependent on meteorological data, since these were used to compute the

foehn index FI that the supervised learning methods in the paper use as truth (response variable). It will therefore not be possible to compute an independent foehn climatology at Altdorf (lines 521-522) and the quality of the classification will by design always be poorer than the one of the foehn index FI.

B.4 **Streamlining**: The manuscript can be shortened by eliminating redundancies and combining parts.

    a) Section 4.1 merely repeats methods already mentioned in the introduction without adding any further information. Section 4 could start with the data and then move on to methods used. Section 5 can be combined with 4 as it also addresses the methodology.

    b) "Case studies" in the results section is a misnomer since they do not refer to a select event. "Feature sets" would be more appropriate. These sets can then be introduced in the data section by combining current subsections 4.2, 4.3 and 4.4. The results can then also be presented in one single confusion matrix, making it much easier for the reader to spot the difference in performance (and simultaneously shortening the paper). However, a fourth feature set needs to be introduced to properly fulfill the purpose of feature set 3 ("case study" in the current version) – see next issue.

B.5 **Feature set 3 inadequately specified**: The stated purpose of having the feature set ("case study") 3 is to evaluate whether adding further measurement stations can compensate for having a shorter training data set. The test period with data that have not been seen by the models in the training phase must be the same for a proper comparison. Especially for a rare event such as foehn with a large interannual variability selecting different and relatively short test periods can lead to considerably different results. Put succinctly: the test period for the feature set with a shorter training period must also be set to 2019–2020 as for the other feature sets. To disentangle the effects of a shorter training period and more stations, respectively, a fourth feature set needs to be introduced that has the shorter training period 2015–2018 and no additional stations.

B.6 **Large fraction of missing data**: The discussion on the limitations of the method (around line 485) correctly mentions a significant amount of periods without data. It would be good to be quantitative – also already when describing the data in section 4 – since this amounts to approximately 1/3 and 1/2 (!) of the time (cf. Tables 5 and 7). The current set up would therefore be unsuitable for any operational use. However, since most of the machine-learning algorithms used rely on aggregating weak learners, setting the methods up in a way that alternate features are used if a particular feature is not available is possible. This is especially easy to achieve for random forests.

B.7 **Section 4.6 Performance metrics**: Comment B.2 was fulfilled to a large part. What is still missing is the statement in the beginning paragraph that all following performance measures are derived from the confusion matrix. It is incorrect to state (line 250) that the performance measures were introduced by Barnes et al. (2007). They have been around many decades before.

B.8 **Percent vs percentage points**: "Percent" is sometimes incorrectly used instead of "percentage points", e.g. line 324. The difference between the observed frequency of foehn in the foehn index – 4.7 % – does not lie within "one percent" (as stated) of the results from the two algorithms. The difference is actually 21 % and 15 %, respectively, which is substantial. What the authors meant is "within one percentage point". However, what is of interest in judging the performance is the *relative* difference of the rare event "foehn", i.e. (correct) percentages

B.9 **Misleading response**: The response to the two points raised in the section "Technical corrections" of my first review simply reads "These have been included in the revised version." This is simply not the case and a lie. Figs. 5, 9, and 10 mentioned in comment C.2 do not appear in any form in the revision any more. Topography (comment C.1) was added but the lines connecting the stations contributing to the top features were omitted.

**C.  Less crucial and technical issues**

C.1 A less crucial item to be changed is: A confusion matrix containing 4 cells does not need a figure. A simple table will do. And if all three (or four, see comments above) feature sets from Figs. 3, 5 and 7 are combined into one table, they can be directly and easily compared.

C.2 And one technical correction: Fig. 2 needs a colorbar.

**References**

Barnes, Lindsey R. et al. (2007). "False Alarms and Close Calls: a Conceptual Model of Warning Accuracy". In: *Weather and Forecasting* 22.5, pp. 1140–1147. DOI: 10.1175/waf1031.1.

Hastie, T, R Tibshirani, and J Friedman (2009). *The Elements of Statistical Learning*. Second Ed. Springer, New York, p. 524. DOI: 10.1007/978-0-387-84858-7.

Mayr, Georg J. et al. (2018). "The Community Foehn Classification Experiment". In: *Bulletin of the American Meteorological Society* 99.11, pp. 2229–2235. DOI: 10.1175/bams-d-17-0200.1.

---

## Referee Report (RR2)

1. $mf_h(e)$、$mf_w(e)$、$mf_g(e)$ should be el in the formula 2, e stands for water vapor pressure.

2. Please describe clearly why beta is set to 2 in formula 8.
3. Please label the abscissa of the upper two graphs in Figure 9.
4. CS1 in label of table 1 should be full named.
5. The training period and test period of CS3 are different from CS1 and CS2, does this affect the evaluation of the model?
6. From line 404 to 405, please cite some papers or explanations that illustrate that GNSS parameters have slightly longer response times to o a change in synoptic conditions.
7. In line 515, whether to consider using other observations to supplement missing tests instead of linear interpolation?

---

## Author Response (AR2)

**Answers to Comments Referee 1**

**General response**

We want to thank referee one for the second extensive review and advice on some points where the manuscript still could/should be improved/corrected. Comments B1 and B5 were very valuable for the last corrections and extensions. We tried to incorporate most of the other comments as well.

We also want to clearly state here that missing some corrections was not done intentionally and the majority of comments on the first version has been implemented.

In the latest version, one figure combining the dropped plots (now Figure 6) was re-introduced to allow for a cross-check. This one includes all foehn events recorded by MeteoSwiss and our predictions (only events where all GNSS data was available) for all four experiments.

In the following, we will go through all specific comments brought up.

**Specific comments**

1. *Correctness of some computations in doubt: The review of the original manuscript version had unearthed large numbers of missed events in the observations of the foehn index FI (even though they are not missing at the MeteoSwiss database as I could confirm; original Fig. 5) and the events diagnosed from GNSS measurements (original Figs. 5–7 and Figs. 9- 10). How the authors "have improved the corresponding code" remains vague in the response. I would have expected to see at least one figure showing observed and diagnosed events in the revised version but these figures have been completely dropped, which eliminates a chance to at least visually inspect the appropriateness of the results. That raises nagging doubts. . .. What is left for a cross-check are the performance metrics. If coding mistakes had been made only in the way missing GNSS data are handled (as the response indicates), differences should be seen in the performance metrics of the test data and the cross-validated training data. However, the numbers for the cross-validated training data metrics in the original and the revised version (Table 2 in both) are identical whereas the metrics for the test data have changed. As far as I see, some computations still have to be wrong. Consequently, the manuscript should not be published unless the correctness of the computations can be convincingly shown.*

   Unfortunately, the scores of the cross-validation have not been updated in the last version: We thank you for cross-checking, this needed to be corrected. The new scores can now be found in the latest version of the manuscript. However, the general outcome has not changed, and the Gradient-Boosting and Support-Vector-Classifier algorithms are still the best performing in terms of combined scores (COMB and F2)

and therefore chosen for the feature setup experiments as we now call them (see point 4). Furthermore, these results did never have any impact on the general conclusion of the paper, which is drawn from the feature setup experiments.

Moreover, we still show observed and diagnosed events, see Figure 9 in the old version (comparison with station data from Altdorf for December 2019). However, as this only represents ~one week of data, we decided to re-introduce comparison plots for diagnosed vs observed events for the experiments. They are now combined into Figure 6 (in the newest version). This hopefully should erase last doubts about the results.

2. ***Selection and application of machine-learning methods: It is important to point out in the paper that no universal truth exists for the diagnosis of foehn and even diagnoses by human experts vary considerably (Mayr et al., 2018). Therefore the classification problem belongs to the category of unsupervised learning (e.g. Hastie et al., 2009). The paper, on the other hand, uses supervised learning methods, a choice that needs justification. A consequence of the use of supervised methods is raised in the following issue.***

We have tried to point this out even more than in the newest version (lines 81-87). How to tackle this problem adequately without supervision is very uncertain to us. Unsupervised learning can cluster/group the data in order to discover something that is not visible otherwise. Considering the highly imbalanced data set and the multitude of phenomena affecting tropospheric delays, it is doubtful that a clear cluster would emerge solely corresponding to foehn events, not to mention foehn at a specific location.

Therefore, it was an obvious choice to apply supervised learning with a reference label (FI index from Dürr (2008)), whose quality has been proven by comparisons to human forecasts (as shown in Dürr (2008)). It is important to note that supervised machine learning is commonly applied to problems with imperfect labels.

Dürr, B.: Automatisiertes Verfahren zur Bestimmung von Föhn in Alpentälern, Arbeitsberichte der MeteoSchweiz, 223, 22 pp, 2008.

3. ***Dependence on foehn classified with traditional meteorological measurements. Despite the claim of the paper (line 445 and lines 521–522) the foehn classification with GNSS data is completely dependent on meteorological data, since these were used to compute. foehn index FI that the supervised learning methods in the paper use as truth (response variable). It will therefore not be possible to compute an independent foehn climatology at Altdorf (lines 521-522) and the quality of the classification will by design always be poorer than the one of the foehn index FI.***

While it is true the method is not fully independent of meteorological data (as not even the estimation of GNSS parameters is completely), the dependence is limited to meteorological data of the training period. Thus, our algorithm(s) is/are dependent exactly on meteorological data from 2010-2018 or 2015-2018, depending on the actual experiment. However, it is not dependent on contemporaneous meteorological data, that is what we would call "completely dependent". Still, we relativized/excluded the benefits of "almost/near-independence" in the latest version, compared to the last

one, as it should be clear to potential readers anyway which data is needed to obtain results.

Furthermore, it might not be possible to compute a fully independent climatology for Altdorf, but for a e.g. 30-year time period excluding 2010/2015-2018, the dependence will be very small. Still, as this might also not be a main use case in the future, we excluded this point from the discussion/conclusion.

In general, we think the fact that for this method (and any supervised learning method) the performance will always be weaker than the "truth" method (here FI index) in direct comparison should actually be quite clear to potential readers of the journal without further explanation. Still, we have added this fact as a point to the limitations/conclusions.

It was also never expected that GNSS measurements will perform better than measures based on actual meteorological data. It is already quite satisfying that the agreement is relatively high.

4. ***Streamlining: The manuscript can be shortened by eliminating redundancies and combining parts.***

   ***a) Section 4.1 merely repeats methods already mentioned in the introduction without adding any further information. Section 4 could start with the data and then move on to methods used. Section 5 can be combined with 4 as it also addresses the methodology.***

   ***b) "Case studies" in the results section is a misnomer since they do not refer to a select event. "Feature sets" would be more appropriate. These sets can then be introduced in the data section by combining current subsections 4.2, 4.3 and 4.4. The results can then also be presented in one single confusion matrix, making it much easier for the reader to spot the difference in performance (and simultaneously shortening the paper). However, a fourth feature set needs to be introduced to properly fulfill the purpose of feature set 3 ("case study" in the current version) – see next issue.***

   a) We tried to combine some of these parts in the new version. Sections 3, 4 and 5 have been combined and rearranged. Section 3 (Methodology as a whole) now starts with datasets, then introduces the default feature setup and finally gives information on the ML algorithms tested and how the two, which were further used, are chosen.

   b) "Case studies" was changed "feature setups", we hope that this suits better.

5. ***Feature set 3 inadequately specified: The stated purpose of having the feature set ("case study") 3 is to evaluate whether adding further measurement stations can compensate for having a shorter training data set. The test period with data that have not been seen by the models in the training phase must be the same for a proper comparison. Especially for a rare event such as foehn with a large interannual variability selecting different and relatively short test periods can lead to considerably different results. Put succinctly: the test period for the feature set with a shorter training period must also be set to 2019–2020 as for the other feature sets. To disentangle the effects of a shorter training period and more stations, respectively, a fourth feature set needs to be introduced that has the shorter training period 2015–2018 and no additional stations.***

Thank you for bringing this point up. It is true that the results of FS1/2 and FS3 (or CS in old version) are not comparable 1:1. We therefore added the requested fourth setup (FS4), which contains the shorter period (2015-2019) but no additional stations (so the station setup of FS1 and FS2). The new results confirm the old ones, i.e. the additional stations bring benefits for the method (see feature importances). However, the decreased number of covered events when new stations are missing must be considered. We also note that 2019 had a higher foehn probability than e.g. 2020, which also explains (some of) the relatively better performance of FS4 compared to FS1.

6. ***Large fraction of missing data: The discussion on the limitations of the method (around line 485) correctly mentions a significant amount of periods without data. It would be good to be quantitative – also already when describing the data in section 4 – since this amounts to approximately 1/3 and 1/2 (!) of the time (cf. Tables 5 and 7). The current set up would therefore be unsuitable for any operational use. However, since most of the machine-learning algorithms used rely on aggregating weak learners, setting the methods up in a way that alternate features are used if a particular feature is not available is possible. This is especially easy to achieve for random forests.***

The large fraction of missing data is for sure the biggest problem of the current version of our method. We have stated this several times and provided detailed statistics on the missing data in the manuscript. As mentioned by you, it is also stated in the discussion of disadvantages/future improvements. We have now added quantitative measures such as percent points of missing events for each FS (in tables describing the setups).

As the aim of this study is a proof-of-concept, it was not the intention to present a method which is ready for operational usage. The large fraction of missing data is for sure the most urgent problem to be dealt with. Nevertheless, if this can be solved, one could even guess (we leave it at this word) that without these missing periods, results might even be better as also more foehn events would show up in the training data.

The issue will anyhow be investigated in further studies, where different solutions for the problem can be tested. Thank you for your proposal of using alternate features, this might be a good way to deal with it. Another easy (but computationally more expensive) possibility is to train a set of different algorithms, iteratively excluding GNSS data from one station in the feature setup each time. This gives (at least) the opportunity to provide a continuous solution as long as only data from one station is missing. We have added both strategies to the outlook section.

7. ***Section 4.6 Performance metrics: Comment B.2 was fulfilled to a large part. What is still missing is the statement in the beginning paragraph that all following performance measures are derived from the confusion matrix. It is incorrect to state (line 250) that the performance measures were introduced by Barnes et al. (2007). They have been around many decades before.***

We have added this statement in the new version (lines 246 and 247).

The sentence (line 250) on the performance measures has been corrected to "see e.g. Barnes et. al (2007)".

8. ***Percent vs percentage points:** "Percent" is sometimes incorrectly used instead of "percentage points", e.g. line 324. The difference between the observed frequency of foehn in the foehn index – 4.7% – does not lie within "one percent" (as stated) of the results from the two algorithms. The difference is actually 21% and 15 %, respectively, which is substantial. What the authors meant is "within one percentage point". However, what is of interest in judging the performance is the relative difference of the rare event "foehn", i.e. (correct) percentages.*

We changed "percent" to "percent points". The relative difference of 21% and 15% might be more substantial, but the result is still quite solid in our opinion.

9. ***Misleading response:** The response to the two points raised in the section "Technical corrections" of my first review simply reads "These have been included in the revised version." This is simply not the case and a lie. Figs. 5, 9, and 10 mentioned in comment C.2 do not appear in any form in the revision any more. Topography (comment C.1) was added but the lines connecting the stations contributing to the top features were omitted.*

We are very sorry to have missed this issue which was due to the fact that the plots have been dropped and the response was not properly updated. However, we also want to clearly state here that all this was definitely not done intentionally.

We have re-introduced the content of Figures 5, 9 and 10 and combined them for the new setups in one figure (Figure 6).

We have also added the lines to the map in the newest version.

**Comments C:**

1.) The confusion matrices of all experiments can now be found in Table 8.
2.) The colorbar (legend) was added to the station map.

**Answers to Comments Referee 2**

**General response**

*Thank you for the revised manuscript, I think it is greatly improved, also regarding the plots. I just have some few final remarks. Let me start with addressing some of the older comments.*

We want to thank the referee for his/her kind words and comments which have significantly contributed to improve the manuscript. The remaining comments and remarks are answered in the following:

1. *Eq.1: I asked the authors in the previous review to change the formula from Rüger and now, it is correct (but the authors asked me what was wrong before):The first two terms were 77.68P/T + 6.3938 e/T When it should be 77.68P/T - 6.3938 e/T (minus instead of plus), because if we substitute P=Pd+e, then we get: 77.68(Pd+e)/T - 6.3938 e/T=77.68Pd/T + 71.2952e/T (the original formulation from Rüger).*
   Thank you very much for the clarification.

2. *"6. Section 4.1: I would recommend giving here at least very brief overview of the selected methods Answer: As the manuscript is quite long already, we actually would like to leave the reader here with the references given describing the methods" -> In my opinion the manuscript is not dramatically long and a brief overview would really help the readers that are not familiar with all the ML methods to get a grasp of the methods the authors are using. I would suggest to at least write something about your two chosen methods, GB and SVC.*

   We have now tried to combine this issue with point number 4, providing a short description of the two used algorithms (GB and SVC) as well as the tuned hyperparameters of them.

3. *Line 163: '(negative) maximum' - > why not use 'minimum' here? Answer: We would interpret 'minimum' as close to zero but one can for sure argue to use minimum here as well" -> just a clarification, in mathematics the local and global minima (and as well a maxima) of a function do not have anything to do with being close to zero but they are simply the largest and smallest values of a function. In here, (for the differences) you have two obvious local minima (out of which one is also the global minimum)*
   Thanks for the clarification, we have changed this.

4. *Line 259: Would be good to comment here what the chosen parameters mean. Answer: Actually, this is done in the caption of (what is now) Table 3" -> I still think it is a good idea to put also an explanation in the main text.*

   Was done in combination with point 2, see above.

*Some new comments:*

1. *Line 130: '1999-2020' – later in the text you state that you only use the data from 2010 or even 2015, so what is a point of mentioning the data from 1999-2010? I*

*understand that it is to point out you have these longer time series but it is confusing for the reader.*

We have changed it to 2010-2020, thanks for pointing this out.

2. *Figure 2: I would make the coordinate font smaller and the stations font larger.*

   The station font was made larger and the coordinate font smaller to improve visibility.

3. *Line 323: Please put dot before 'Figure'. In general, I would suggest to check the punctuation, there are some commas missing and some too many (e.g. Line 327: 'Furthermore it can be seen, that' -> 'Furthermore, it can be seen that' etc.)*

   We have gone through the manuscript once again and updated/corrected everything we found.

4. *Line 375, 378: remove 'again' – you do not show the same again, these are new plots*

   Requested by referee one, we decided to drop the plots of the confusion matrices and instead give their entries summarized in Table 10. Therefore, also these lines have been dropped/modified.

**Answers to Comments Referee Report 3**

**General response**

We want to thank referee three for the comments on how to further improve it. All specific comments are addressed below.

**Specific comments**

**1. $mf_h(e)$、 $mf_w(e)$、 $mf_g(e)$ should be el in the formula 2, e stands for water vapor**

Thanks for having a close look here. We have updated the formula.

**2. Please describe clearly why beta is set to 2 in formula 8.**

Beta is a real number factor which determines the weighting of recall vs. precision (or vice versa) in the F-score. As already mentioned in the manuscript, for our (highly imbalanced) classification problem recall is a better representation of accuracy than precision. This is because optimal performance in terms of precision would/could result in the best performing algorithm never predicting foehn, as it is such a rare phenomenon. Thus, we use the F2 score (beta = 2) which weights the recall parameter two times larger than precision.  We added some more description to the manuscript.

**3. Please label the abscissa of the upper two graphs in Figure 9**

We have added the appropriate label DOY to the figure (now Figure 7).

**4. CS1 in label of table 1 should be full named**

The label has been updated as requested, but keep in mind that (due to the request of referee 1) it now states FS1, for "Feature setup".

**5. The training period and test period of CS3 are different from CS 1 and CS 2 does this affect the evaluation of the model?**

The short answer to this is simply yes, it does. Choosing e.g. 2020 as the test period will yield different results to a certain extent. However, our main intention for this setup was to still apply the 80/20 training/test data rule-of-thumb as also done for the other case studies/experiments. We added a fourth feature set (station setup from CS1 and 2 for the reduced period) to achieve a fair comparison with the extended station setup. This was requested by referee one.

**6. From line 404 to 405 please cite some papers or explanations that illustrate that GNSS parameters have slightly longer response times to o a change in synoptic conditions**

What was meant here was actually the fact that different GNSS stations (and their absolute values) might experience a change in synoptic conditions later (or actually sooner depending on their location). Therefore, this was not formulated correctly and updated in the newest version. We are sorry for confusing here.

However, also the temporal resolution, which the tropospheric parameters estimated, might also play a role in cases of rapid changes in synoptic conditions. As correlation of the tropospheric parameters with other station parameters requires longer observation periods, e.g. to separate the zenith total delay from the height component or the east-west gradient from longitude, a stacking period of 15 min is recommended for tropospheric parameter

estimation [Dach et. al, 2015]. In recent works, the temporal resolution is further reduced to 1 min [e.g., Hadas and Hobinger, 2021] using a Kalman filter approach. Therefore, additional relative constrains are introduced leading to slightly longer response times in case of rapid changes in synoptic conditions.

Still, as we have hourly values of both troposphere estimates and foehn index, this might not be a huge problem for our results.

Dach, R., S. Lutz, P. Walser, P. Fridez (Eds); 2015: Bernese GNSS Software Version 5.2. User manual, Astronomical Institute, University of Bern, Bern Open Publishing. DOI: 10.7892/boris.72297; ISBN: 978-3-906813-05-9.

Hadas T., Hobiger T.: Benefits of Using Galileo for Real-Time GNSS Meteorology. IEEE Geoscience and Remote Sensing Letters, Vol. No. , Piscataway, NJ, USA 2020, pp. 1-5. DOI: 10.1109/LGRS.2020.3007138

**7. In line 515, whether to consider using other observations to supplement missing tests instead of linear interpolation?**

This depends on what is meant with "other observations".

Using additional meteorological measurements would most likely help to improve the performance of the method. However, this is not our intention as the goal is to rely only on GNSS-based observations.

Using observations from different GNSS stations (which are currently not part of the feature setup) for filling gaps should be possible but takes a more sophisticated setup of the ML algorithm and a more detailed analysis of their impact. This might be a topic addressed in a follow-up study. Another possibility which we might also investigate in the future (and also added to the conclusions/outlook section) is to build a larger set of algorithms, each one excluding one GNSS station from the feature setup. This could provide a continuous solution as long as not more than one station is missing.

**Author's changes in manuscript (from revised version)**

**Main changes:**

- **Updated scores of cross-validation/algorithm selection:**
  We changed the scores of the cross-validation to those achieved by the new study setup. These changes were unfortunately missed to update in the revised version. Although the changes are minor and not influencing the following results and conclusion, we thank referee one for pointing towards this issue.
  Furthermore, there are small changes in some of the statistics of the three FS experiments due to:
    1.) A bug that was encountered in the calculation of POFD and MAR, which has now been corrected for all experiments. Therefore, these values (slightly) changed for all results shown.
    2.) The fact that we re-ran all the experiments using the newest sklearn-version (1.1.2), in order to cross-check if there are any changes when using this version. Slight changes in some cases occur, which are now updated in the manuscript and in the model code/software provided. Thus, the 1.1.2 version of sklearn should be used when using the provided code.

  However, these changes have not influenced significantly the conclusions of the study.

- **Addition of a fourth experiment:**
  As requested by referee one, we added a fourth experiment (or feature setup) to enable a better comparison with FS3, which uses a different training/test setup (2015-2018/2019) compared to the first two experiments (2010-2018/2019-2020). Feature setup 4 (FS4) now contains the station setup of FS1 and FS2 but the training/test split of FS3. Therefore, the comparison between FS3 and FS4 now directly addresses the impact of additional stations on the results.

- **Re-introduction of prediction time series plots for comparison:**
  As requested by referee one, we re-introduced plots to compare results from the GB/SVC algorithms with observed events (FI from MeteoSwiss) over the whole test period for the different experiments. All four of those lots have been combined into what now is Figure 6. As stated in the caption of Figure 6, please note that this now includes all observed events, so also those were no GNSS data is available and thus no prediction could be made.

- **"Case study" has been changed to "Feature setup":**
  As requested by referee one, we changed "case study" to "feature setup".

- **Sections combined:**
  As requested by referee one, we combined former sections 3, 4 and 5 under "Methodology". These parts have also been slightly rearranged. The section now starts with data sets, then introduces the default feature/station setup and finally presents the ML algorithm cross-validation and descriptions of the two chosen algorithms.

**Some of the minor changes:**

- Short descriptions of the Gradient-Boosting and Support-Vector-Classifier algorithms were added (as requested by referee two)
- Lines connecting the five top predictors (ZWD differences) were added to Figure 2 as requested by referee one. We are sorry to have missed this issue in the first revision. Furthermore, the station font was increased as suggested by referee two and a colorbar (what we would interpret as a legend here) was added.